# Ground Based Hyperspectral Imaging to Characterize Canopy-Level Photosynthetic Activities

**Yu Jiang** [1,2,†] **, John L. Snider** [3,†] **, Changying Li** [1,*] **, Glen C. Rains** [4] **and Andrew H. Paterson** [3,5,6]

1   School of Electrical and Computer Engineering, College of Engineering, The University of Georgia, Athens, GA 30602, USA; yujiang@uga.edu

2   Horticulture Section, School of Integrative Plant Science, Cornell AgriTech, Cornell University, Geneva, NY 14456, USA; yj522@cornell.edu

3   Department of Crop & Soil Sciences, College of Agricultural & Environmental Sciences, The University of Georgia, Tifton, GA 31793, USA; jlsnider@uga.edu (J.L.S.); paterson@uga.edu (A.H.P.)

4   Department of Entomology, College of Agricultural & Environmental Sciences, The University of Georgia, Tifton, GA 31793, USA; grains@uga.edu

5   Department of Genetics, Franklin College of Arts and Sciences, The University of Georgia, Athens, GA 30602, USA

6   Department of Plant Biology, Franklin College of Arts and Sciences, The University of Georgia, Athens, GA 30602, USA

*   Correspondence: cyli@uga.edu

†   These authors contributed equally to this work.

**Abstract:** Improving plant photosynthesis provides the best possibility for increasing crop yield potential, which is considered a crucial effort for global food security. Chlorophyll fluorescence is an important indicator for the study of plant photosynthesis. Previous studies have intensively examined the use of spectrometer, airborne, and spaceborne spectral data to retrieve solar induced fluorescence (SIF) for estimating gross primary productivity and carbon fixation. None of the methods, however, had a spatial resolution and a scanning throughput suitable for applications at the canopy and sub-canopy levels, thereby limiting photosynthesis analysis for breeding programs and genetics/genomics studies. The goal of this study was to develop a hyperspectral imaging approach to characterize plant photosynthesis at the canopy level. An experimental field was planted with two cotton cultivars that received two different treatments (control and herbicide treated), with each cultivar-treatment combination having eight replicate 10 m plots. A ground mobile sensing system (GPhenoVision) was configured with a hyperspectral module consisting of a spectrometer and a hyperspectral camera that covered the spectral range from 400 to 1000 nm with a spectral sampling resolution of 2 nm. The system acquired downwelling irradiance spectra from the spectrometer and reflected radiance spectral images from the hyperspectral camera. On the day after 24 h of the DCMU (3-(3,4-dichlorophenyl)-1,1-dimethylurea) application, the system was used to conduct six data collection trials in the experiment field from 08:00 to 18:00 with an interval of two hours. A data processing pipeline was developed to measure SIF using the irradiance and radiance spectral data. Diurnal SIF measurements were used to estimate the effective quantum yield and electron transport rate, deriving rapid light curves (RLCs) to characterize photosynthetic efficiency at the group and plot levels. Experimental results showed that the effective quantum yields estimated by the developed method highly correlated with those measured by a pulse amplitude modulation (PAM) fluorometer. In addition, RLC characteristics calculated using the developed method showed similar statistical trends with those derived using the PAM data. Both the RLC and PAM data agreed with destructive growth analyses. This suggests that the developed method can be used as an effective tool for future breeding programs and genetics/genomics studies to characterize plant photosynthesis at the canopy level.

**Keywords:** solar induced fluorescence; cotton; maximal fluorescence estimation; growth analysis

## 1. Introduction

The global population is likely to exceed 10 billion by 2050, presenting great challenges for agriculture [1]. To fulfill the needs of the rapidly growing population, the current agricultural yield must be doubled by that time, which translates into an annual increase of 1.75% total factor productivity (TFP) [2]. Currently, the global TFP growth rate is approximately 1.5%, leaving a gap of 0.25% annually. Even worse, the TFP growth rate is only approximately 0.96% in developing countries, which is far behind the required growth rate. Cotton (*Gossypium*) is one of the most important textile fibers in the world, accounting for about 25% of total world textile fiber use [3]. Thus, improvement of cotton production is vital for fulfilling the fiber requirements of over ten billion people by 2050 [4].

As with all agricultural crops that have reproductive structures of economic importance, the yield of cotton can be expressed as a function of total seasonal light interception, radiation use efficiency, and harvest index [5]. Thus, yield improvement can be achieved by increasing any one of these three variables. The "Green Revolution" introduced dwarfing genes into the most important C3 cereal crops (e.g., rice and wheat), allowing an increased biomass allocation to grain with a reduction in the total aboveground biomass (thus, an increased harvest index) [6]. Breeding programs have continued to increase carbon allocation into grain [7], and in cotton specifically, genetic yield improvement historically has also been associated with an increase in biomass partitioning to reproductive units (bolls) [8,9]. One study [10], however, suggested that future yield improvements in high potential environments would likely be achieved by (1) selecting cotton varieties that exhibit a more indeterminate growth habit (i.e., capitalize on the high insolation levels experienced in long growing season environments); and (2) increasing photosynthetic efficiency either through breeding or biotechnology efforts. Photosynthesis is a process that converts radiant energy into biochemical energy, and is the basis of plant growth. In contrast with efforts to breed for desirable plant growth habits or greater harvest index, photosynthetic improvement has not yet been achieved for breeding programs, and remains a promising avenue for increasing agricultural productivity in the future. In addition, variations in photosynthetic efficiency can be used as indicators of plant stress, which can be used for selecting genotypes with high levels of stress tolerance or for making management decisions at the field scale [11]. Furthermore, there is a tremendous amount of interest in using remote sensing to model gross primary productivity of natural ecosystems according to the original framework of [5,12]. In order to achieve this, the photosynthetic efficiency of the canopy must be estimated.

Chlorophyll fluorescence parameters are often used to evaluate photosynthetic performance and stress in plants [13]. These parameters include three measurable variables; i.e., minimal fluorescence ($F_0$ or $F_0'$) when photosystem II (PSII) centers are open, maximal fluorescence ($F_m$ or $F_m'$) when PSII centers are closed, and steady state fluorescence ($F_s$ or $F_s'$). There are several derived variables, such as variable fluorescence ($F_v = F_m - F_0$ or $F_v' = F_m' - F_0'$) and difference in fluorescence ($F_q = F_m - F_s$ or $F_q' = F_m' - F_s'$) between $F_m$ (or $F_m'$) and $F_s$ (or $F_s'$). Variables denoted by primes are for light-adapted states, and are otherwise for dark-adapted states. These parameters can be used to calculate the maximum ($\frac{F_v}{F_m}$ or $\frac{F_v'}{F_m'}$) and operating ($\frac{F_q'}{F_m'}$, also known as, $\Phi_{PSII}$) efficiencies of PSII photochemistry, both of which are useful for photosynthetic efficiency evaluation and plant stress detection [14].

There are active and passive sensing approaches to measure chlorophyll fluorescence parameters. Active techniques include pulse amplitude modulation (PAM) [15] and laser induced fluorescence transience (LIFT) [16], both of which can emit predefined light to measure minimal fluorescence ($F_0$), steady state fluorescence ($F_s$), and maximal fluorescence ($F_m$), and can calculate variable fluorescence ($F_v$). Based on active techniques, portable instruments (e.g., PAM fluorometers) have been developed

and widely used for photosynthesis studies [17–22]. Active techniques rely on artificial illumination, however, which limits their use for large scale applications [23].

Passive techniques retrieve chlorophyll fluorescence emission excited by solar illumination (natural sunlight), which is termed solar induced fluorescence or sun induced fluorescence (SIF). Upwelling radiance from plants under solar illumination is a mixture of SIF and surface reflectance, and it is feasible to decouple SIF signals from the upwelling radiance in Fraunhofer lines of the solar spectrum in which irradiance is substantially reduced because of atmospheric absorption (e.g., hydrogen and oxygen). In the red and far-red spectral range, three Fraunhofer lines are frequently used for SIF retrieval, including *Hα* at 656 nm because of hydrogen and $O_2$-*B* at 687 nm and $O_2$-*A* at 760 nm because of oxygen. Common SIF retrieval approaches include conventional Fraunhofer line discrimination (FLD), improved FLD variants, and reflectance-based ratios [24]. Many studies explored the use of ground based spectrometers to retrieve the SIF values of plant leaves, and reported high correlations between SIF measurements and fluorescence measured using active techniques (e.g., PAM fluorometers) [24–27]. Large national research institutions (e.g., the European Space Agency, the EPA, and the National Aeronautics and Space Administration (NASA)) launched programs to investigate the use of hyperspectral imagery sensed remotely from planes and satellites to monitor SIF changes at the regional and global levels, to estimate carbon fixation (or flux of carbon dioxide) and gross primary productivity (GPP). Multiple studies have shown strong correlations between gross primary productivity (whole canopy photosynthesis) and SIF through various modeling methods [12,28,29]. When using SIF to track GPP, there is usually a strong association with the absorbed photosynthetically active radiation (PAR). In situations where light intensity is constant and extreme stress (such as heat stress) limits canopy photosynthesis, however, the relationship is somewhat degraded, because excess energy that cannot be dissipated through non-photochemical quenching (NPQ) might be emitted as increased fluorescence [30]. Thus, it is important to develop a method to generate canopy level quantum yields and photosynthetic activities if the method can be broadly applicable under a range of environmental conditions. These data would be invaluable for research studies and policy-making to secure the food supply [28,29,31–33].

Spectrometer-based approaches can provide the highest accuracy of measurement location (a specific point on a leaf), but have an obvious limitation in the scanning throughput (point by point). By contrast, airborne and spaceborne solutions provide a substantially faster scanning throughput, but have compromised spatial resolutions (sub-meter to meters). Neither approach, therefore, would be suitable for breeding programs and plant-science studies at the canopy or sub-canopy levels. Ground-based hyperspectral imaging would be a viable solution to address those issues because it provides a higher scanning throughput (usually line by line) and a better spatial resolution (sub meter or higher). Many researchers have explored the use of ground-based hyperspectral imaging for extracting vegetative indices that can be correlated with plant aboveground biomass [34], leaf area index [35], and various plant pigments (e.g., chlorophyll) [36]. Although these studies showed some success, they required either sophisticated model calibration and validation [36] or artificial illumination [34]. A few studies examined the use of ground hyperspectral imaging to measure SIF, showing high measurement accuracies and the potential of spatial variation analysis [37,38]. However, these studies were restricted to the scope of instrumentation validation and did not utilize retrieved SIF for characterizing whole-canopy photosynthesis.

It should be noted that passive hyperspectral sensing systems can also measure canopy reflectance indices to monitor photosynthetic activities. For instance, the photochemical reflectance index (PRI) was introduced by Gamon et al. [39], and has been studied extensively [40–42]. It is calculated as $(R_{531} - R_{570})/(R_{531} + R_{570})$ and has been correlated with photosynthetic efficiencies previously. However, some studies reported that the PRI is somewhat limited, in that measurements are highly sensitive to viewing angle and are strongly influenced by soil background at leaf area index (LAI) values less than 3 [12,41]. In particular, a previous study found no correlation between PRI and light

use efficiency (LUE) [43]. Thus, it would be of great interest to focus on the measurement of SIF rather than other reflectance indices for photosynthetic activity evaluations, especially at the canopy level.

A particularly notable limitation to using SIF to estimate canopy-level photosynthetic efficiency is the method for estimation of maximal fluorescence ($F'_m$) at the canopy level, which has not been reported previously [44]. If $F'_m$ could be determined for the canopy using SIF, real-time estimates of crop-level photosynthetic efficiencies could be obtained using passive sensing approaches. Direct measurement of $F'_m$ faces a significant challenge: the maximum intensity of solar illumination on the ground is far less than a "saturating" flash intensity used by PAM fluorometers, so maximal fluorescence at the canopy level cannot be directly measured. In fact, it has been demonstrated that leaves of plants previously acclimated to high light environments often do not close all reaction centers despite exposure to such a "saturating" flash intensity, preventing even direct measurement of $F'_m$ by using PAM fluorometry. A multi-phase flash approach was developed, however, to estimate $F'_m$ and electron transport rate (ETR) without the "saturating" light [45]. Using this method, a leaf sample is successively exposed to flash intensities in either ascending or descending order, and the intensity of chlorophyll fluorescence is quantified at each step. Fluorescence intensity ($F$) is then plotted against the reciprocal of PAR. A function is fit to the data to obtain the y intercept and an estimate of apparent $F'_m$ at an infinite light intensity when all reaction centers would hypothetically be closed. Since it is often necessary to expose single leaf samples to different PAR levels in succession to obtain an estimate of $F'_m$, it might also be possible to estimate $F'_m$ for the canopy by taking advantage of natural, diurnal variations in PAR and SIF.

The overall goal of this study was to develop a ground-based hyperspectral imaging approach to characterize photosynthesis at the canopy level. Specific objectives were to (1) develop a ground-based hyperspectral imaging system to measure diurnal SIF changes using the FLD method; (2) calculate effective quantum yield and ETR to derive light curves for characterizing plant photosynthesis; and (3) validate the efficacy of the characterization method by comparing the hyperspectral imaging derived measurements with PAM-derived measurements and destructive crop growth analysis.

## 2. Materials and Methods

### 2.1. Plant Materials and Experimental Design

To evaluate the utility of diurnal SIF measurements for estimating whole canopy photosynthetic efficiency, a field experiment was established at the University of Georgia Lang-Rigdon research farm near Tifton, GA. Two cotton cultivars (PHY 841 RF and PHY 340 W3FE) were planted on 19 June, 2018 at a 2.5 cm depth, with a seeding rate of 11 seeds per m$^2$. Individual plots were 1 row with a length of 9 m and an inter-row spacing of 0.91 m. Plots were arranged along the east–west direction. The two cultivars planted represent two different species of cotton adapted to different cotton production regions of the southern United States (US) and were previously shown to differ in leaf anatomical characteristics and photosynthetic response to light intensity [46]. PHY 841 RF is a Pima cotton (*Gossypium barbadense*) cultivar widely grown in the arid southwestern US, whereas PHY 340 W3FE is an upland cotton (*Gossypium hirsutum*) cultivar primarily grown in the humid southeastern US. To generate large differences in photosynthetic efficiency of the canopy, once the crop had reached the initial stages of floral bud development (referred to as "squaring"), plots received one of two possible herbicide treatments. Untreated control plots received only water applied as a foliar spray at a rate of 130 L/ha. Diuron (3-(3,4-dichlorophenyl)-1,1-dimethylurea) treated plots had a 41% solution of DCMU (*w/w*; Diuron 4L) applied at a rate of 2.35 L/ha and a total application volume (water plus Diuron solution) of 130 L/ha. Diuron is a highly selective herbicide that specifically blocks the transfer of electrons from PSII to plastoquinone during the thylakoid reactions of photosynthesis. There were two rows of buffer plants between adjacent herbicide treatments to prevent drift onto non-target plants. Fertility, pest control, and irrigation were managed according to University of Georgia's Cooperative Extension recommendations [47]. The experiment was arranged as a split plot,

randomized complete block design, wherein DCMU treatment represented the whole plot factor and cultivar represented the sub-plot factor. There were eight replicate plots for each cultivar within a particular herbicide treatment.

### 2.2. Data Acquisition

#### 2.2.1. Hyperspectral Data Acquisition

The GPhenoVision system was configured with a spectral module for spectral data acquisition [48]. The spectral module consisted of one spectrometer (Flame VIS-NIR, Ocean Optics Inc., Largo, FL, USA) and one hyperspectral camera (MSV500, Middleton Spectral Vision, Middleton, WI, USA) (Figure 1). Both sensors were calibrated radiometrically and spectrally (also spatially for the hyperspectral camera), covering the spectral range from 400 nm to 1000 nm with a spectral sampling resolution of 2 nm. For the spectral calibration, the spectrometer was calibrated by the manufacturer, whereas the hyperspectral camera was calibrated by the authors using three calibration lamps. Details of spectral and spatial calibration can be found in a previous study [37]. For the radiometric calibration, the spectrometer, the hyperspectral camera, a spectroradiometer (Ocean Optics Inc., Largo, FL, USA), and an illumination source (DC-950, Fiber-Lite, Dolan-Jenner Industries, Boxborough, MA, USA) were attached to an integrating sphere (4P-GPS-060-SF, Labsphere, North Sutton, NH, USA). By adjusting the illumination intensity at different levels, calibration models between digital counts and irradiance were established for the spectrometer and hyperspectral camera. The spectrometer was equipped with a cosine corrector (field of view of 180°) facing towards the sky, whereas the hyperspectral camera was positioned nadir to the ground. The spectrometer acquired the irradiance spectra of the sunlight. Depending on the solar irradiance intensity, the sampling frequency of the spectrometer varied from 20 Hz to 50 Hz, so that signal intensities could stay in the optimal range without saturation. The hyperspectral camera was positioned 2.15 m above the ground, collecting radiance spectral images of plant canopies of two plots at a time. To ensure the spatial aspect ratio, the hyperspectral camera ran at 100 frames per second (FPS) and the platform moved at an approximate speed of 0.5 m/s. The system moved sunward along the plot direction (east-west direction) to avoid potential issues caused by shading effects. Six data collection trials were conducted on 9 August 2018 (beginning approximately 24 h after Diuron application) at 08:00, 10:00, 12:00, 14:00, 16:00, and 18:00, respectively.

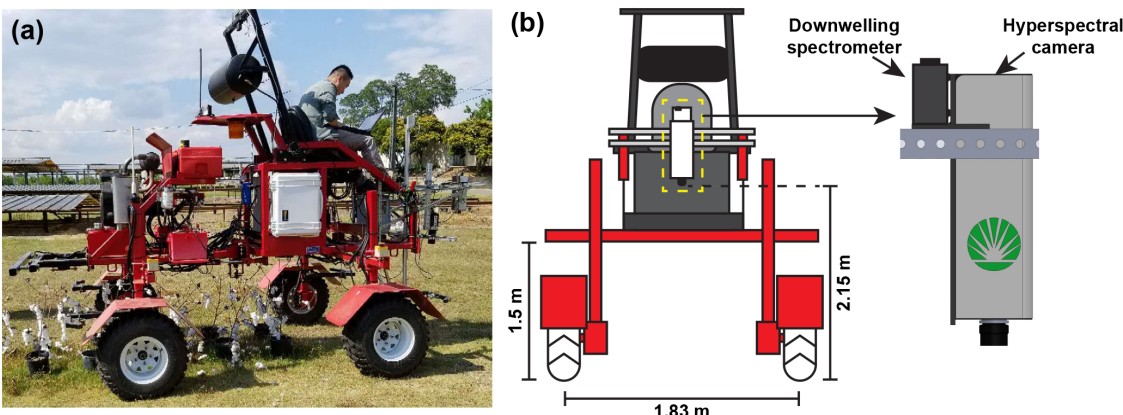

**Figure 1.** Illustration of data collection system: (**a**) picture of the "GPhenoVision" system and (**b**) diagram of the system configuration and sensor installation.

#### 2.2.2. Fluorometry Measurement

Active chlorophyll fluorescence measurements were conducted simultaneously with hyperspectral imaging collection. A human operator followed the GPhenoVision system and used a portable pulse-amplitude-modulation (PAM) fluorometer (OS5p+, Opti-Sciences, Inc., Hudson, NH, USA) to measure the uppermost fully expanded leaf at approximately the fourth mainstem node below

the plant terminal. Three leaves per plot were measured. At each diurnal sampling time, the leaf blade was clipped so that the orientation of the exposed adaxial surface relative to incoming solar radiation was left unchanged, and steady state fluorescence ($F_s'$) was measured under ambient light conditions using a 660 nm modulation measuring beam under naturally occurring solar irradiance as our actinic light source. While measuring fluorescence, PAR at the leaf surface was estimated using a PAR sensor integrated into the leaf clip. Subsequently, maximal fluorescence intensity ($F_m'$) was estimated using a multi-phase flash (provided by a 35 W halogen bulb) approach comparable to the methods described in [45], wherein relative fluorescence intensity is plotted versus the reciprocal of PAR following exposure of the leaf sample to a sequence of flashes with increasing intensity (2850, 5700, and 8550 μ mol/m$^2$/s) for a total duration of 0.95 s. A linear function was fit to the resulting data set to estimate $F_m'$ at infinite light intensity. This represents fluorescence intensity when all reaction centers are closed.

*2.3. Characterization of Canopy-Level Photosynthetic Efficiency*

2.3.1. Retrieval of Solar Induced Fluorescence

Collected spectral data were used to retrieve SIF values at the canopy level (Figure 2). Irradiance spectra of the sunlight and radiance hyperspectral cubes of the plant canopy were synchronized using timestamps, resulting in meta-hyperspectral cubes of each scanning row, where individual pixels had both irradiance and radiance spectra. Based on the spatial information, a meta-hyperspectral cube was further split into two sub-cubes, with each sub-cube containing irradiance and radiance data for one plot.

The following processes were performed for the meta hyperspectral cube of a single plot. Based on preliminary tests, grayscale images at 749 and 685 nm were used to generate a band ratio image ($I_{749}/I_{685}$). A threshold was applied to the band ratio image to create the mask of plant canopies in that plot. An arbitrary value of 3 was used in the present study based on the trial-and-error method. Chlorophyll absorbs incident light (particularly blue and red light in the visible spectral range) and emits fluorescence in the red and far-red spectral range. Upwelling radiance spectra of plant canopies thus contain both reflectance and fluorescence signals in the red and far-red spectral range. Fraunhofer lines are a set of spectral absorption lines in the spectrum of sunlight related to particles in the solar and terrestrial atmosphere. *Hα* at 656 nm, $O_2$-*B* at 687 nm, and $O_2$-*A* at 760 nm are three Fraunhofer lines in the red and far-red spectral range. The reduction of solar irradiance in the Fraunhofer lines results in a decrease of the canopy reflectance and an increase in the ratio of fluorescence and reflectance signals, which maximizes the suitability of decoupling chlorophyll fluorescence from the canopy reflectance. A standard Fraunhofer line discrimination (sFLD) method has been developed to use irradiance and radiance signals at two spectral bands [49]. One band is one Fraunhofer line, and the other band is a wavelength near the corresponding Fraunhofer line. In the present study, based on previous literature review [24], the $O_2$-*A* band (approximately at 761 nm) and its neighboring band (759 nm) were used to calculate SIF values of individual pixels using Equation (1). It is noteworthy that the sFLD method was used, because the present study focused on exploring the possibility of using hyperspectral imaging for SIF retrieval and successive canopy-level photosynthetic analysis. Other Fraunhofer line discrimination models (e.g., 3FLD and improved FLD) can be used to improve SIF retrieval accuracy in future studies [24].

$$SIF_p = \frac{E_p^{759} L_p^{761} - E_p^{761} L_p^{759}}{E_p^{759} - E_p^{761}},$$
(1)

where $SIF_p$ was the SIF value (W/m$^2$/nm/sr) of a pixel $p$. $E_p^{\cdot}$ and $L_p^{\cdot}$ represented the irradiance (W/m$^2$/nm) and radiance (W/m$^2$/nm/sr) intensities of $p$ at a certain wavelength.

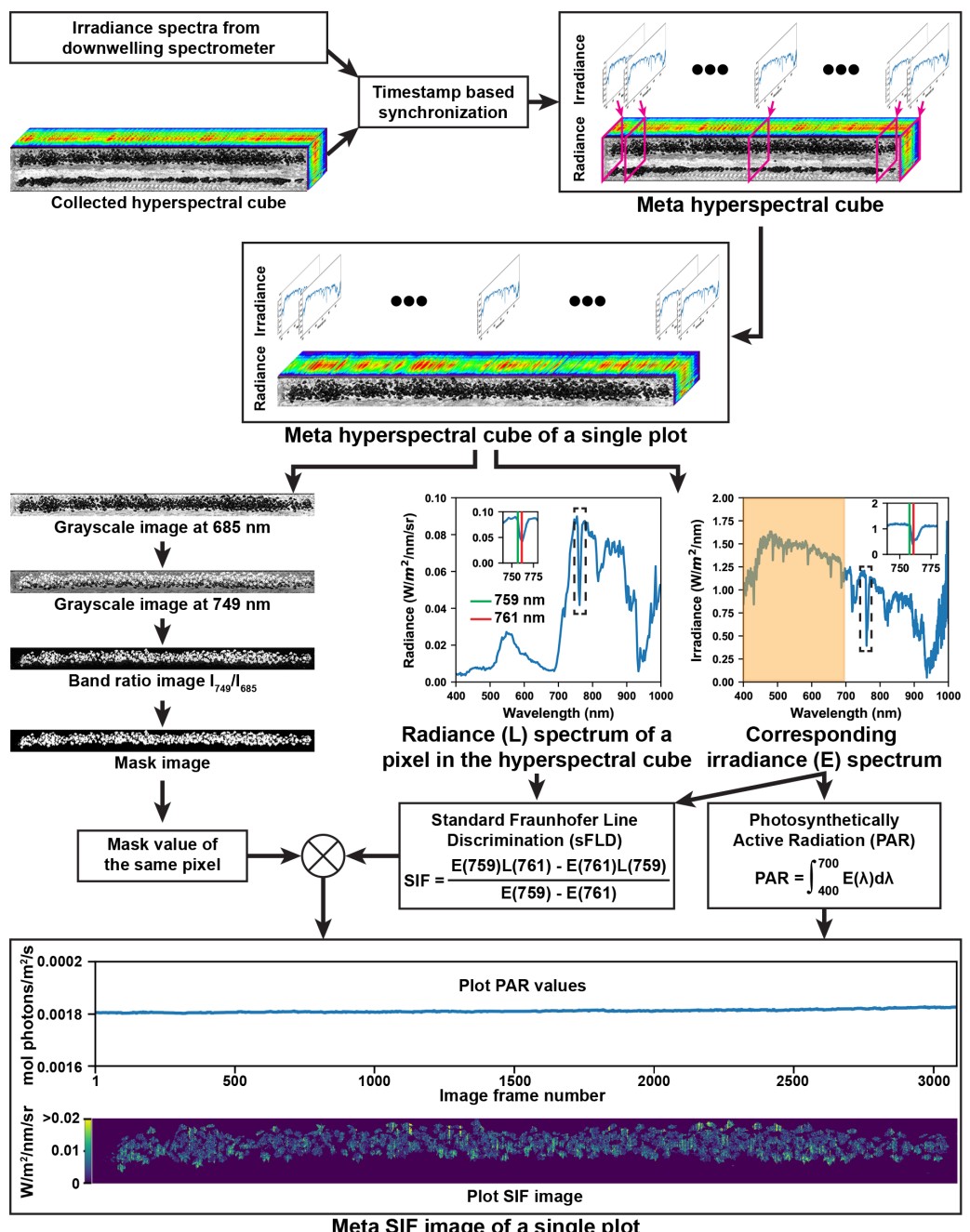

**Figure 2.** Flowchart of image processing from collected raw data to meta-SIF (solar induced fluorescence) images. In the meta-hyperspectral cube, each pixel had both irradiance and radiance spectra covering the spectral range from 400 to 1000 nm with a spectral resolution of 2 nm.

The plant canopy mask was multiplied with retrieved SIF values, forming the SIF image of the plot. As each image line of a radiance hyperspectral cube had a corresponding irradiance spectrum, PAR values were calculated for individual lines of SIF images using Equation (2). Pixels along individual lines in a radiance hyperspectral cube shared the same irradiance spectrum measured by the spectrometer.

$$PAR_p = \frac{1}{A_s t_s} \int_{400}^{700} \frac{E_p^\lambda}{e_\lambda} d\lambda = \frac{1}{A_s t_s} \int_{400}^{700} \frac{E_p^\lambda \lambda}{hc} d\lambda, \tag{2}$$

where $PAR_p$ was the PAR value (μ mol/m$^2$/s) of a pixel $p$. $A_s$ was the surface area (m$^2$) of the cosine corrector equipped with the spectrometer, and $t_s$ was the integration time (s) of the spectrometer. $E_p^\lambda$

was the irradiance of $p$ at the wavelength $\lambda$ (*nm*). $e_\lambda$ was the energy of a photon at the wavelength $\lambda$, $h$ was the Planck constant, and $c$ was the light speed in a vacuum. One PAR value corresponded to the average value for all pixels in one line of a hyperspectral cube.

Subsequently, a meta-SIF image was generated for a plot, consisting of a plot SIF image and a PAR curve (PAR values for lines along the row direction of the corresponding SIF image).

### 2.3.2. Calculation of Effective Quantum Yield and Electron Transport Rate

For data derived from hyperspectral images (HSI), the potential maximum SIF value was estimated for each of four combinations of genotype and treatment. In each genotype-treatment combination, a regression model was used to fit SIF measurements over reciprocals of PAR, and the model's y-interception (the reciprocal of PAR equaled to zero) was treated as the maximum SIF value for all plants in that combination. An exponential model was used for the two control treatment combinations, whereas three different models were used for the two Diuron combinations due to a lack of obvious data distribution patterns (likely due to damage to the photosynthetic apparatus and resulting non-photochemical quenching). The three models for the Diuron combinations included a linear regression model, quadratic regression model, and average model that used the mean value of SIF measurements as the maximum SIF. After obtaining the maximum SIF value for each approach, the effective quantum yield of PSII ($\phi_{PSII}$) and photosynthetic ETR were calculated using Equations (3) and (4) for data points in each combination.

$$\phi_{PSII} = \frac{SIF_m - SIF}{SIF_m} = 1 - \frac{SIF}{SIF_m} \tag{3}$$

$$ETR = PAR \times \phi_{PSII} \times A_{leaf} \times R_{PSII}, \tag{4}$$

where $\phi_{PSII}$ was the effective quantum yield of PSII for a SIF value ($SIF$). $SIF_m$ was the estimated maximum SIF value for a genotype-treatment combination. $A_{leaf}$ was the leaf absorbance of incident light, and a typical value (0.84) for C3 species (e.g., cotton) was used in this study [50,51]. $R_{PSII}$ was the distribution ratio of absorbed energy between photosystem I (PSI) and photosystem II (PSII), which was assumed as an equal distribution (0.5) [52].

For PAM fluorometry data, the maximal fluorescence intensity and PAR were obtained, so $\phi_{PSII}$ was calculated using Equation (5) [53], while ETR was still calculated using Equation (4).

$$\phi_{PSII} = \frac{F_m' - F_s'}{F_m'} = 1 - \frac{F_s'}{F_m'}, \tag{5}$$

where $\phi_{PSII}$ was the effective quantum yield of PSII. $F_s'$ and $F_m'$ were the steady state and maximal fluorescence intensities under actinic light.

### 2.3.3. Rapid Light Curve and Standardized ETR

Because light intensity influences electron transport rates and can fluctuate throughout the day or even within the same measurement period for a given plot, rapid light response curves were generated from diurnal data to provide a standardized measure of maximum electron transport rate for each plot. ETR and PAR values were used to generate rapid light curves (RLCs) using the method proposed by [54]. To quantitatively analyze RLCs, RLCs were fitted using an empirical model (Equation (6)) [55].

$$ETR = mETR \times (1 - e^{-\alpha \times PAR/mETR}), \tag{6}$$

where $mETR$ was the photosynthetic capacity at saturating light by which all reaction centers were hypothetically closed, and $\alpha$ was the initial slope of the RLCs before the onset of saturation; i.e., the slope of the light-limiting region of RLCs.

RLCs were generated and modeled at two levels: the group level and the plot level. RLCs at the group level were calculated using all data points ($N$ = 48) of a group (a genotype-treatment combination), and RLC parameters were compared to evaluate differences in canopy-level photosynthetic efficiency among the four groups. In order to test the statistical significance of these differences, RLCs at the plot level were calculated using data points ($n$ = 6) of individual plots in each group. Consequently, an individual group had 8 replicates for conducting statistical analyses.

While *mETR* can be used to compare differences of photosynthetic efficiency between groups, standardized ETR (sETR) values were calculated by setting PAR equal to 1500 $\mu$ mol/m$^2$/s, which is a common value in the study area and is generally considered a saturating light intensity for cotton [56].

### 2.4. Calculation of Standardized Photochemical Reflectance Index

Photochemical reflectance index (PRI) values were computed from collected hyperspectral images for individual plots at different sampling time periods. For each plot, a logarithmic model ($PRI = a \times ln(PAR) + b$) was established between diurnal PRI and PAR. To compare the potential of using PRI and SIF for characterization of photosynthetic activities, standardized PRI (sPRI) values were calculated by setting the same PAR (1500 $\mu$ mol/m$^2$/s) used for sETR calculation.

### 2.5. Growth Analysis

Crop performance was also verified by destructively harvesting all aboveground plant material in a 2 m long section of each row on two sampling dates to derive classical crop growth indices. Plants were sampled on 9 August 2018 (immediately after SIF measurements) and 23 August 2019 (a two week interval). On each sampling date, plants were placed in plastic bags with moist paper towels to ensure that plant tissues did not desiccate between harvest and measurement. In the laboratory, plants were separated into leaves and stems, and leaf area was determined using a leaf area meter (LI-3100, LI-COR Corp., Lincoln, NE, USA). Total dry weight was assessed following a 48 h drying period at 80 $^\circ$C in a forced-air oven.

Five growth parameters were calculated, including crop growth rate (CGR), net assimilation rate (NAR), relative growth rate (RGR), the difference in leaf area index ($\Delta$LAI), and the difference in leaf mass fraction ($\Delta$LMF) between the two sampling dates. The five parameters were defined by Equations (7)–(11).

$$CGR = \frac{W_{t2}^{total} - W_{t1}^{total}}{(t2 - t1) \times A^{land}} \tag{7}$$

$$NAR = \frac{W_{t2}^{total} - W_{t1}^{total}}{t2 - t1} \times \frac{\ln A_{t2}^{leaf} - \ln A_{t1}^{leaf}}{A_{t2}^{leaf} - A_{t1}^{leaf}} \tag{8}$$

$$RGR = \frac{\ln W_{t2}^{total} - \ln W_{t1}^{total}}{t2 - t1} \tag{9}$$

$$\Delta LAI = \frac{A_{t2}^{leaf} - A_{t1}^{leaf}}{A^{land}} \tag{10}$$

$$\Delta LMF = \frac{W_{t2}^{leaf}}{W_{t2}^{total}} - \frac{W_{t1}^{leaf}}{W_{t1}^{total}}, \tag{11}$$

where $W^{leaf}$ and $W^{total}$ represented the leaf and total dry weights. $A^{leaf}$ and $A^{land}$ represented the leaf and land areas. In the present study, the $A^{land}$ was 1.82 m $^2$ (0.91 m $\times$ 2 m). $t2$ and $t1$ were the sampling dates in days after planting (DAPs).

### 2.6. Statistical Analysis

Least squares linear regression analyses were performed to calculate the correlation between HSI- and PAM-derived $\phi_{PSII}$ measurements, which evaluated the goodness of different methods

for maximal fluorescence estimation. To avoid potential effects due to outliers, these analyses were conducted using robust regression option ("bisquare") in MATLAB (The MathWorks, Inc., Natick, MA, USA). To test the effectiveness of standardized ETR values, ANOVA tests were performed on the five growth parameters and standardized ETR values estimated using four approaches (PAM and three HSI-based methods). After testing the effects due to genotype, treatment, and the interaction between genotype and treatment, ANOVA tests were further performed on the traits between treatments for each genotype. ANOVA tests were performed at the significance level of 0.05 in R [57]. In addition to ANOVA tests, Pearson correlation analysis was conducted between growth traits, sETR, and sPRI, evaluating the potential of using the sETR for growth prediction. Pearson correlation analyses were also conducted in R.

## 3. Results

### 3.1. Representative Meta-SIF Images

Meta-SIF images showed obviously different trends between the control and Diuron-treated groups (Figure 3). In the control groups, SIF values had the same trend as PAR values: SIF values increased with the increase of PAR values and decreased with the reduction of PAR values. In contrast, SIF values for Diuron-treated plots exhibited no relation with PAR values: SIF values were low and relatively constant, irrespective of PAR changes throughout the day. This observation agreed with the experimental design. The control groups were healthy, showing fluorescence intensity changes along with varied PAR levels, whereas an inhibitor of electron transport beyond PSII would be expected to cause damage to the photosynthetic apparatus and potentially increase the NPQ of the fluorescence signal.

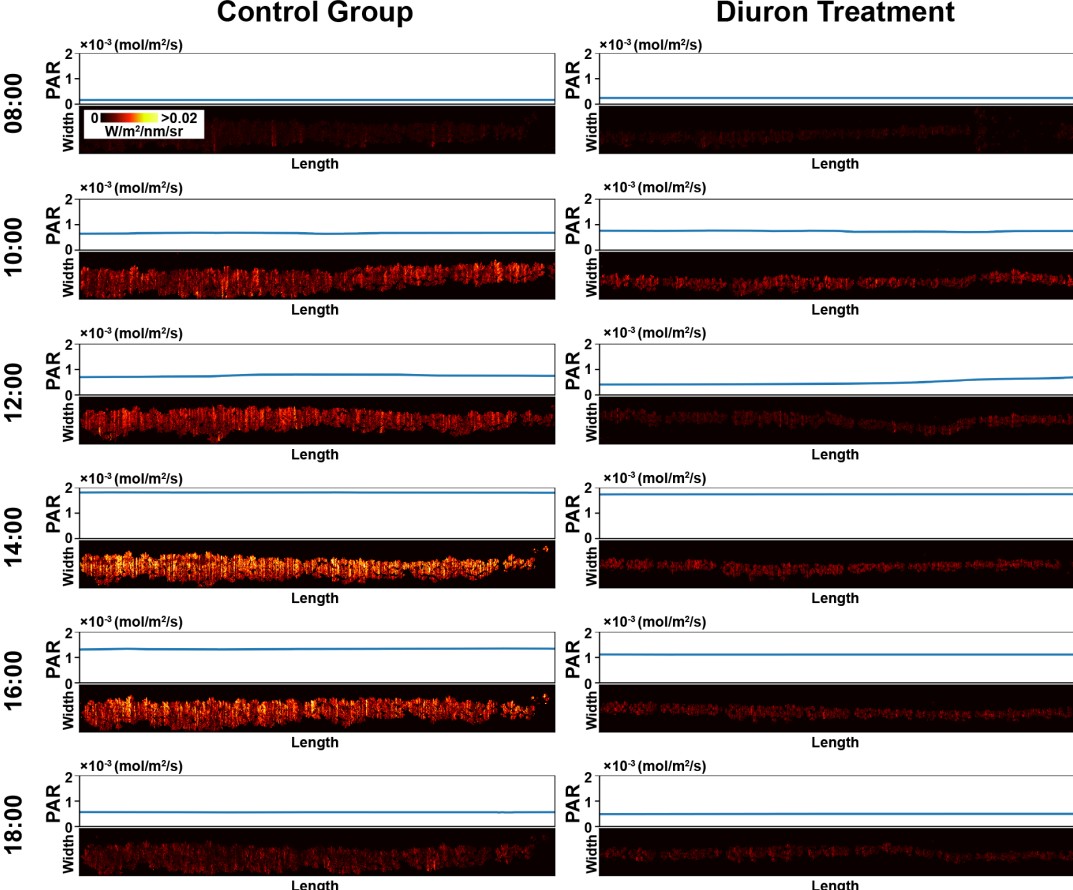

**Figure 3.** Representative meta-SIF images for control and Diuron-treated plots. In meta-SIF images, each pixel had both an SIF value and a corresponding photosynthetically active radiation (PAR) value.

In addition, variations were observed within a plot (see the control group at 1400 h in Figure 3). Leafy regions in the plot showed higher SIF signals than less leafy regions, because the leafy regions usually had a faster vegetative growth (and thus more mature leaves), resulting in a higher capacity for photosynthesis. This suggests that calculated meta-SIF images could be used for analyzing the spatial variation of photosynthesis. As the present study aimed to explore the possibility of using SIF measured by passive HSI method to characterize photosynthetic efficiencies at the canopy level, experiments were not designed to analyze spatial variations of SIF in a plot. Thus, successive studies are needed to take the full advantage of HSI data to analyze spatial variations of SIF (and potentially other photosynthetic parameters).

### 3.2. Estimated Maximal Fluorescence

Generally, the control groups were well ($R^2$ = 0.85) fitted by the exponential model, whereas the Diuron-treated groups showed large variations in the model that best fit the SIF response to PAR (Figure 4). As exponential models showed a strong relationship between SIF and PAR in control groups, it was reasonable to use the y-intercepts of the models as the maximal fluorescence intensities. In contrast, the three models for the Diuron–treated groups provided different goodness-of-fit and maximal fluorescence values. Regarding goodness-of-fit, the quadratic model provided the best estimation of the maximal fluorescence, followed by the linear model and average model. This was possibly because the level of NPQ in Diuron-treated plots would have been higher under the highest light intensities where the most damage would have been expected. Although a previous study showed a linear relationship between chlorophyll fluorescence and PAR under high light intensities (over 2800 $\mu$ mol/m$^2$/s) [45], no strong linear relation was observed between SIF and PAR under solar illumination (up to 2000 $\mu$ mol/m$^2$/s) in the present study. In particular, the linear model provided a reduced goodness-of-fit than the quadratic model, suggesting that the linear model might not be optimal for estimating the maximal fluorescence for the Diuron-treated groups. However, there was a reasonable consideration for using the average model, despite the worst goodness-of-fit. No obvious trend was identified between SIF and PAR values in the Diuron-treated groups, and thus variations among data points could be considered random measurement errors. Granting that, it was acceptable to use the mean value as the group measurement (and thus the maximal fluorescence of that group), thereby reducing the measurement error. Nonetheless, the three models provided different values for maximal fluorescence intensity, and would have had different effects on the successive data processing.

### 3.3. Calculated Effective Quantum Yield and RLCs

Overall, the Pima and Upland cultivars showed the similar trends and patterns for $\phi_{PSII}$ and ETR values calculated using PAM and hyperspectral data (Figure 5). For all four methods, $\phi_{PSII}$ values decreased with increasing PAR values in the range from 0 to 2000 $\mu$ mol/m$^2$/s for control groups, whereas $\phi_{PSII}$ values were relatively lower and showed no correlation with PAR values for Diuron-treated groups. Control and Diuron-treated groups were distinctively different from each other in $\phi_{PSII}$ over PAR calculated using PAM data. While control and Diuron-treated groups were still separable, overlaps between the two groups were identified with different magnitudes in $\phi_{PSII}$ over PAR calculated using HSI data. The overlaps between control and Diuron-treated groups for quantum yield were the largest when $\phi_{PSII}$ values were calculated using maximal fluorescence estimated by the linear model, and the overlaps became smaller, being minimal when $\phi_{PSII}$ values were calculated using maximal fluorescence estimated by the average and quadratic models, respectively. This occurred primarily because of the differences between estimated maximal fluorescence values for the Diuron-treated groups. The HSI quadratic model provided the lowest value of the estimated maximal fluorescence, resulting in the lowest $\phi_{PSII}$ values. In particular, a lower maximal fluorescence value led to more negative $\phi_{PSII}$ values that were treated as zero, which increased the magnitude of the differences between the control and the Diuron-treated groups.

　　　　ETR values rapidly increased with increasing PAR values in the range from 0 to 600 $\mu$ mol/m²/s, and gradually reached a plateau afterwards for control groups. This trend held true for ETR values calculated using PAM- and HSI-derived $\phi_{PSII}$. ETR values calculated using PAM-derived $\phi_{PSII}$ remained low (near zero) in the PAR range from 0 to 2000 $\mu$ mol/m²/s for Diuron-treated groups, showing a clear separation from control groups. In contrast, ETR values calculated using HSI-derived $\phi_{PSII}$ showed different magnitudes of overlap between control and Diuron-treated groups. In the PAR range from 0 to 1500 $\mu$ mol/m²/s, all three HSI-based methods showed a distinction between control and Diuron-treated groups, with the largest overlap by the HSI linear method followed by average and quadratic methods. In the PAR range from 1500 to 2000 $\mu$ mol/m²/s, the HSI linear method showed no distinction between two treatment groups, whereas the HSI quadratic and average methods still showed a distinct difference. This matched with the observations of $\phi_{PSII}$.

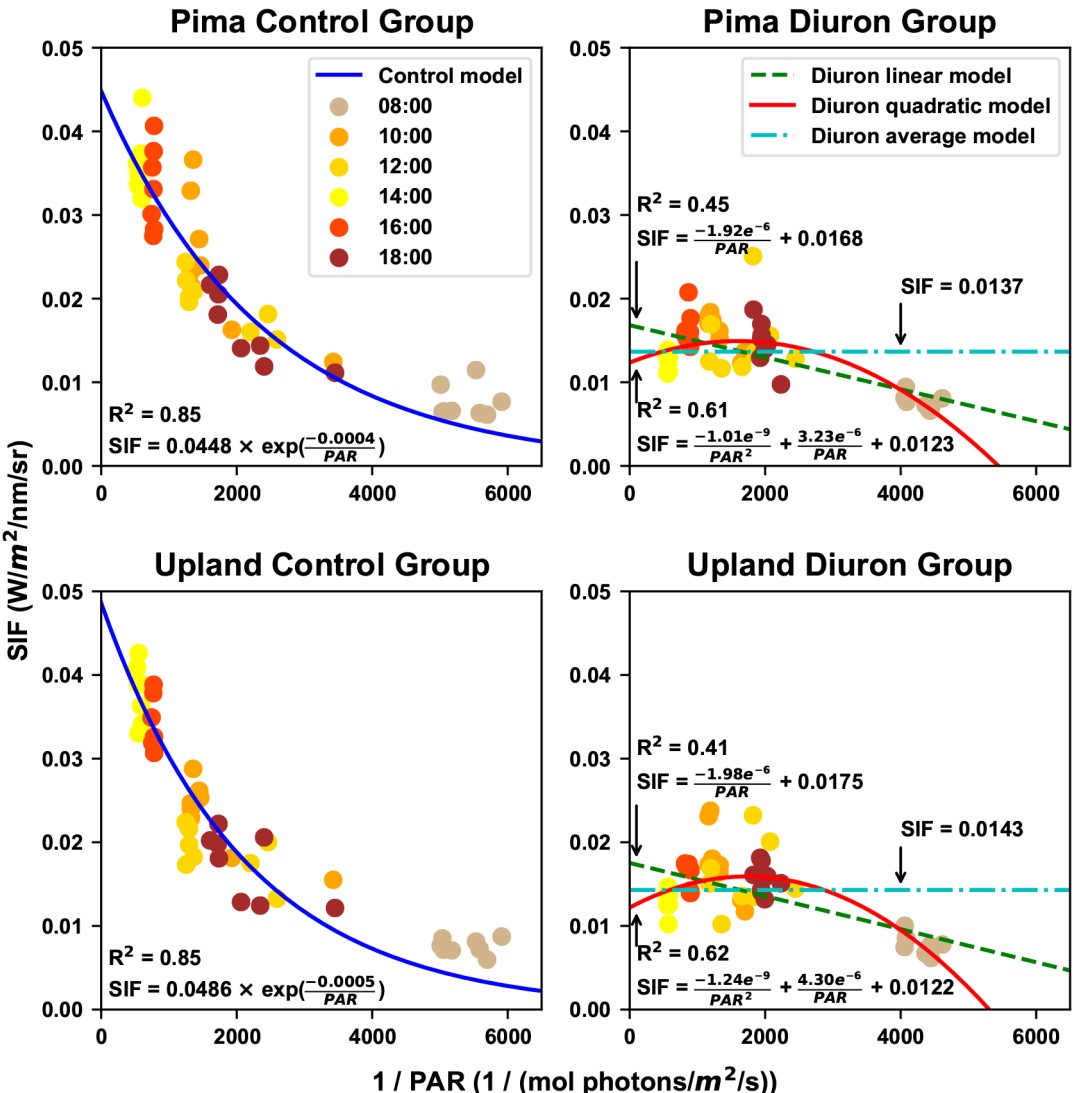

**Figure 4.** Estimation of maximal fluorescence values for control and Diuron treatments. An exponential model was used to fit the SIF and PAR reciprocal values for plots in the control group, whereas three models (linear, quadratic, and average) were used to fit the SIF and PAR reciprocal values for plots in the Diuron-treated group. The maximal fluorescence value was defined as the value when the PAR reciprocal equaled zero.

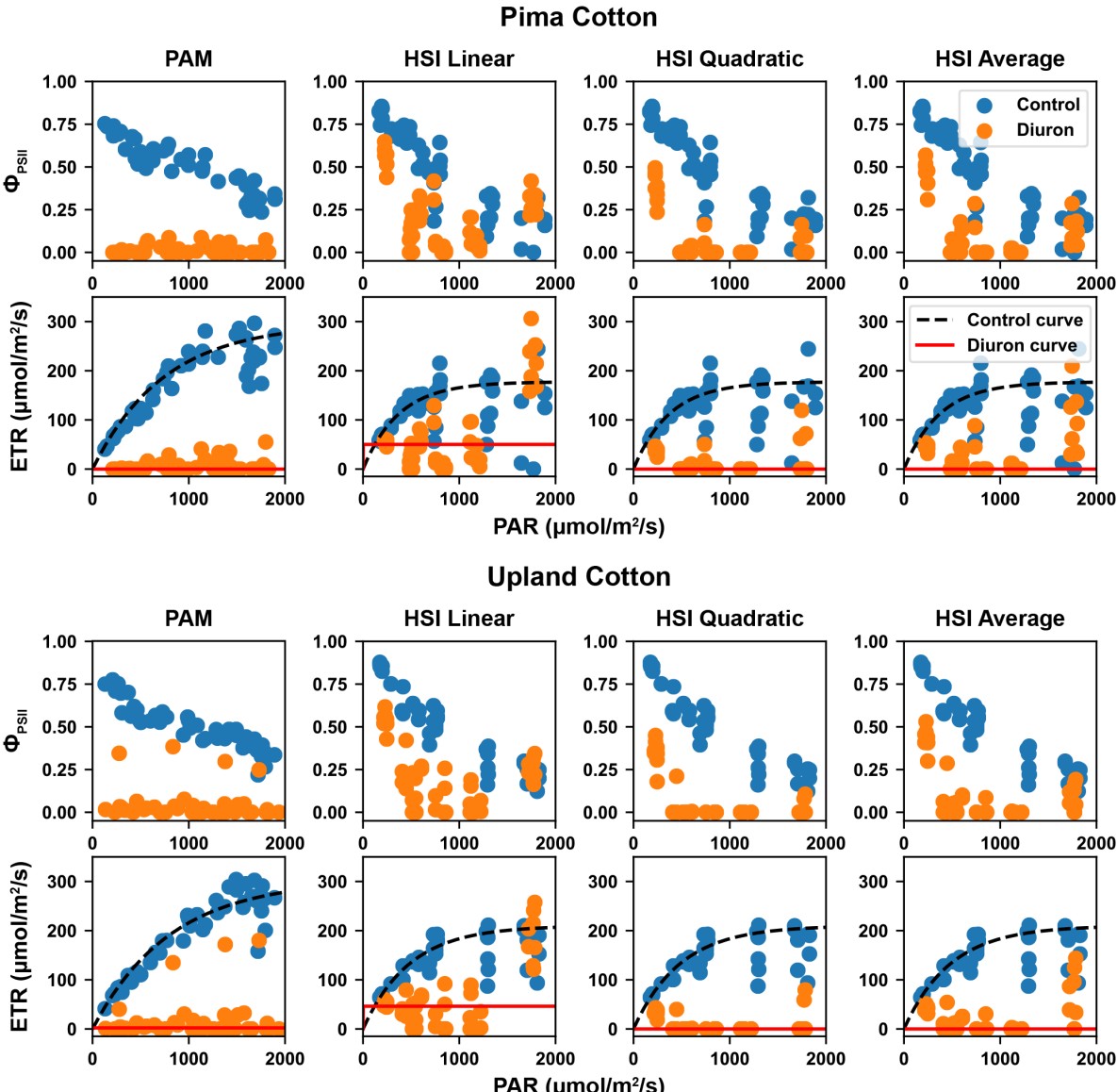

**Figure 5.** Calculated effective quantum yield of PSII ($\phi_{PSII}$) and electron transport rate (ETR) for the two genotypes under control and Diuron treatments. PAR is short for photosynthetically active radiation.

Quantitative RLC models further validated the aforementioned observations (Table 1). For all four methods, control groups had a substantially higher value of $mETR$ and a lower value of initial slope ($\alpha$) than Diuron-treated groups for both cultivars. Although the HSI linear method provided ETR values that had a large overlap between control and Diuron-treated groups, the quantitative characteristics of the fitted RLC models were dramatically different from each other, showing that even in the worst case (HSI linear method), the developed method could be used to identify the photosynthetic efficiency difference between the two groups. Although different methods showed a similar trend in the RLC characteristics between different treatments, the absolute values of those characteristics were different. $mETR$ values calculated using HSI methods were 60% to 70% lower than those calculated using the PAM method, and $\alpha$ values were 15% to 20% higher. The reduction of absolute values for mETR was partially because the PAM and HSI methods provided measurements at different levels. The PAM method measured a single leaf near the top of a plant with little or no shading, whereas the HSI methods measured a whole canopy that had leaves at different levels of shading. So, it would be expected that the HSI methods provided a lower ETR than the PAM method. In addition to herbicide treatment differences, the two cultivars showed certain differences. The Pima cultivar had a lower

minimum saturating irradiance (estimated by $\frac{mETR}{\alpha}$ [55]) than the Upland cultivar. This suggests that the Pima cultivar would enter into the stage dominated by non-photochemical quenching at lower light intensities than the Upland cultivar [58]. This finding is in agreement with a previous study showing that individual leaves of Pima cotton reached light saturation for net photosynthesis at a lower light intensity than Upland cotton [46].

**Table 1.** Rapid light curves (RLCs) calculated using ETR and PAR values for each genotype-treatment combination. The RLC model was $ETR = mETR \times (1 - e^{\alpha \times PAR/mETR})$.

| Method | Cultivar | Treatment | mETR | $\alpha$ | $E_k$ | R2 |
|---|---|---|---|---|---|---|
| PAM | Pima | Control | 296.70 | 0.3989 | 743.73 | 0.94 |
| PAM | Pima | Diuron | 0.00 | 0.5000 | 0.00 | |
| PAM | Upland | Control | 303.90 | 0.3771 | 805.98 | 0.99 |
| PAM | Upland | Diuron | 2.12 | 0.5000 | 4.24 | |
| HSI_Linear | Pima | Control | 177.51 | 0.4780 | 371.33 | 0.92 |
| HSI_Linear | Pima | Diuron | 50.13 | 77.0580 | 0.65 | |
| HSI_Linear | Upland | Control | 210.07 | 0.4348 | 483.13 | 0.99 |
| HSI_Linear | Upland | Diuron | 46.19 | 93.9418 | 0.49 | |
| HSI_Quadratic | Pima | Control | 177.51 | 0.4780 | 371.33 | 0.92 |
| HSI_Quadratic | Pima | Diuron | 0.00 | 0.5000 | 0.00 | |
| HSI_Quadratic | Upland | Control | 210.07 | 0.4348 | 483.13 | 0.99 |
| HSI_Quadratic | Upland | Diuron | 0.00 | 0.5000 | 0.00 | |
| HSI_Avg | Pima | Control | 177.51 | 0.4780 | 371.33 | 0.92 |
| HSI_Avg | Pima | Diuron | 0.00 | 95.0250 | 0.00 | |
| HSI_Avg | Upland | Control | 210.07 | 0.4348 | 483.13 | 0.99 |
| HSI_Avg | Upland | Diuron | 0.00 | 0.5000 | 0.00 | |

$\phi_{PSII}$ measured using different HSI methods also showed different correlations with those measured using the PAM method (Figure 6). In fact, a high correlation ($R^2 = 0.73$) was achieved between PAM and HSI methods for control groups. Differences in correlations mainly came from the measurements for Diuron-treated groups. The quadratic model provided the best estimate of maximal fluorescence, exhibiting the strongest correlation between PAM and HSI-based estimates of quantum efficiency at the canopy level. This suggests that the quadratic model could be an optimal method for estimating maximal fluorescence for Diuron-treated groups. The average model showed a reduced but comparable performance with the quadratic model. Considering its engineering rationale, the average model could also be an option for estimating maximal fluorescence for Diuron-treated groups. Additional experiments are needed to determine which method (the quadratic or average model) would be the most optimal in terms of model generalization capability to different datasets. However, the linear model provided the lowest correlation, suggesting that the linear model would not be suitable for maximal fluorescence estimation for Diuron-treated groups.

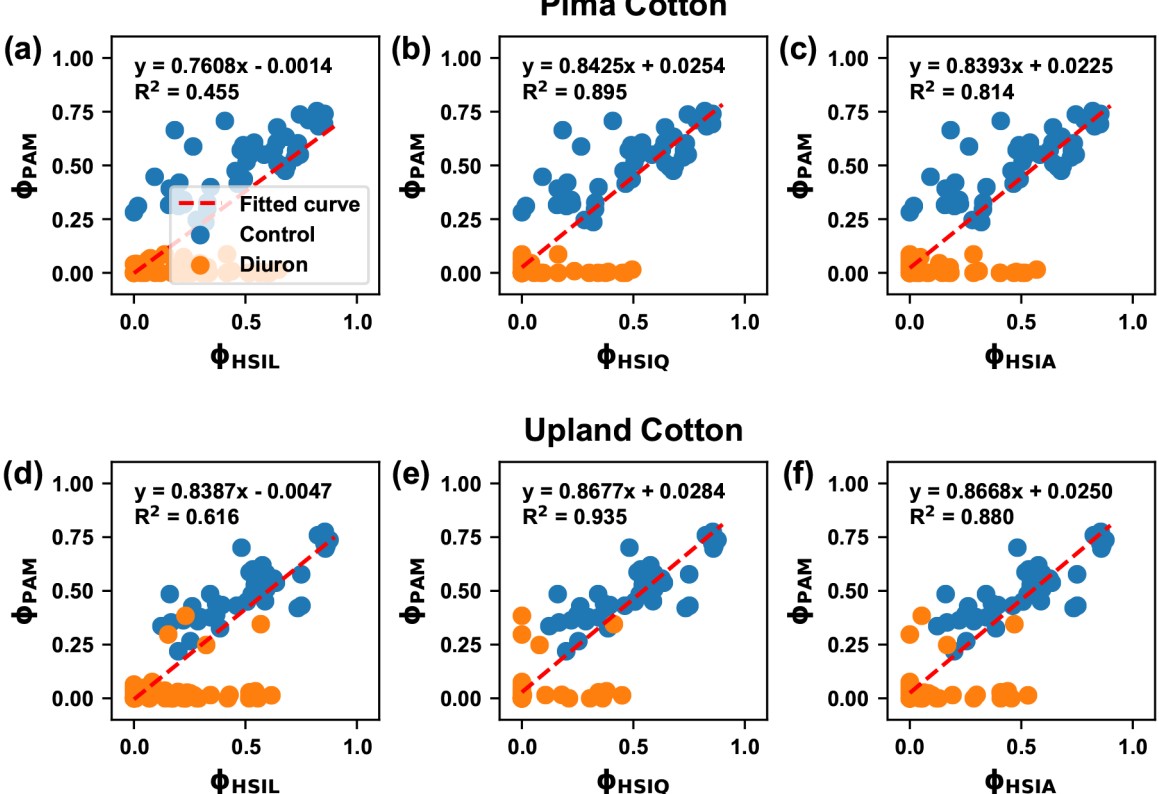

**Figure 6.** Coefficient of determination ($R^2$) between the effective quantum yield of PSII ($\phi_{PSII}$) values derived from PAM and HSI data: (**a–c**) are for the Pima cultivar and (**d–f**) are for the Upland cultivar. $\phi_{PAM}$ indicates values calculated using PAM data. $\phi_{HSIL}$, $\phi_{HSIQ}$, and $\phi_{HSIA}$ indicate values derived from hyperspectral data using the maximal fluorescence value estimated by the linear, quadratic, and average models. It is noteworthy that to avoid potential effects due to outliers, all $R^2$ values were obtained from least-squares linear regression analyses with robust option ("bisquare") in MATLAB.

*3.4. ANOVA Test Results*

Generally, growth traits and sETR calculated using four methods showed significant differences between treatments but no difference between genotypes (Table 2). This suggests that the sETR measurements were effective in identifying the differences in plant growth between control and Diuron-treated groups. It should be noted that although an interaction effect was identified for $\Delta LMF$ and sETR calculated using the HSI methods, control groups still showed higher values than Diuron-treated groups irrespective of genotype, indicating the validity of main effect (treatment) significance for those traits. RGR was the only trait showing a significant difference between genotypes.

For each cultivar, growth traits and sETR calculated using four methods had statistically higher values for control groups than Diuron-treated groups (Figure 7). Growth traits (e.g., CGR, NAR, RGR, and $\Delta LMF$) were positive for control groups, indicating normal plant growth, whereas those traits were close to zero or even negative, indicating plant loss of mass and leaf area in Diuron-treated plots. The same patterns were observed for sETR calculated using four methods. An exception was $\Delta LMF$, which represents dry matter partitioning to leaf area. Both cultivars showed negative $\Delta LMF$ for control and Diuron-treated groups, meaning both cultivars distributed a smaller fraction of total dry matter to leaves at the later sampling times irrespective of treatment. The Pima cultivar exhibited a greater decline in LMF for the Diuron-treated group than the control group, likely reflecting the damage caused by Diuron treatment and resulting defoliation that led to leaf mass fraction reduction. In contrast, the Upland cultivar showed no difference of $\Delta LMF$ between treatments. This is likely because the total growth was negatively affected to a comparable extent as leaf mass.

**Table 2.** The *p*-values of ANOVA tests on the growth traits, and standardized ETR values estimated using four approaches.

| Trait | Cultivar | Treatment | Interaction between Cultivar and Treatment |
| --- | --- | --- | --- |
| CGR | 0.1568 | 0.0001 | 0.8406 |
| NAR | 0.1229 | 0.0002 | 0.3513 |
| RGR | 0.0128 | 0 | 0.2899 |
| $\Delta LAI$ | 0.1801 | 0.0018 | 0.3906 |
| $\Delta LMF$ | 0.9069 | 0.0008 | 0.006 |
| $sETR_{PAM}$ | 0.0792 | 0 | 0.65 |
| $sETR_{HSIL}$ | 0.2661 | 0 | 0.0237 |
| $sETR_{HSIQ}$ | 0.0864 | 0 | 0.0411 |
| $sETR_{HSIA}$ | 0.1663 | 0 | 0.0241 |

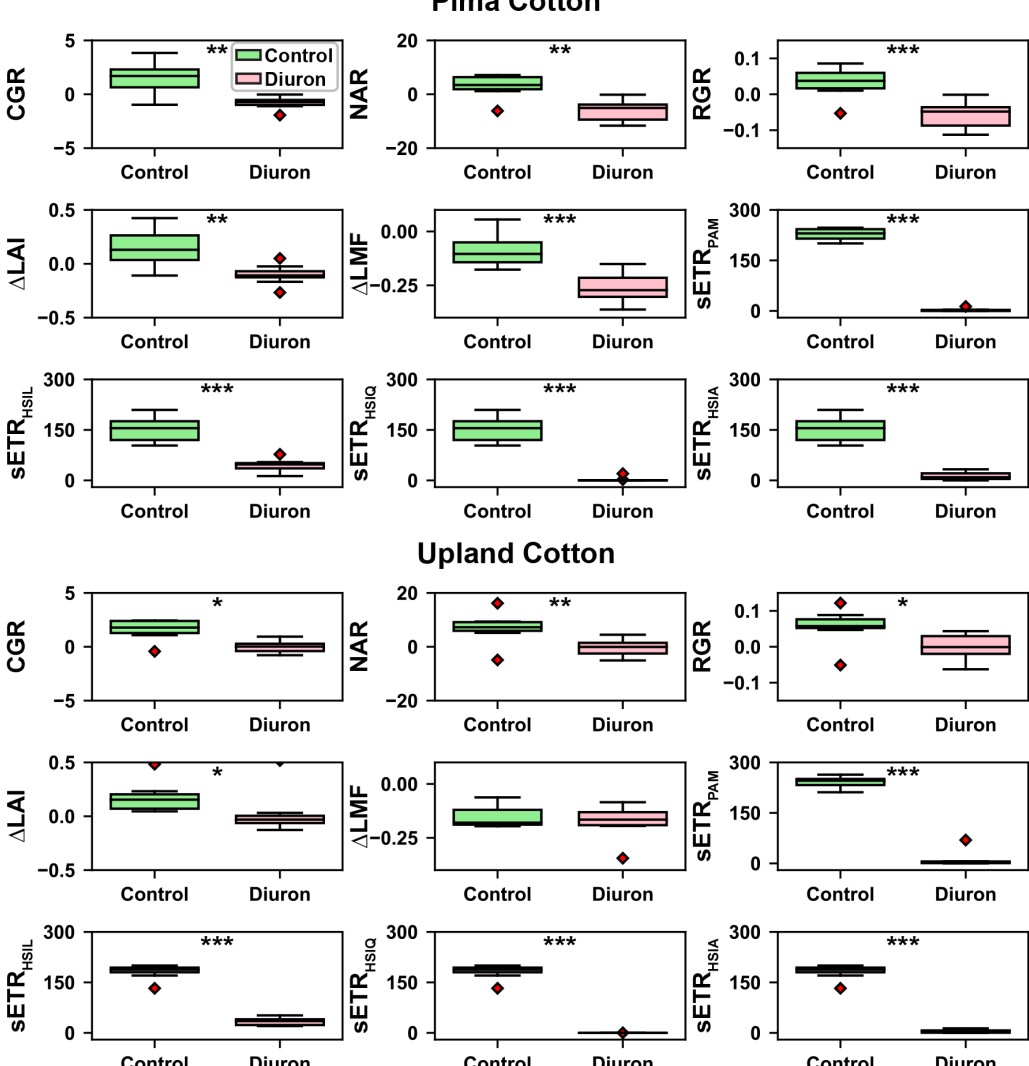

**Figure 7.** ANOVA tests on the growth traits and standardized ETR values estimated using four approaches. Asterisks indicate statistical differences between two treatments at different significance levels: * < 0.05, ** < 0.01, and *** < 0.001. CGR, NAR, RGR, and sETR values are in the units of $g/m^2/d$, $g/m^2/d$, $g/g/d$, and $\mu mol/m^2/s$, respectively.

## 3.5. Correlation between Traits

High correlations (r = 0.92 to 0.95) were achieved for sETR values calculated using the PAM and HSI methods, further indicating the validity of using the developed method for canopy-level

photosynthetic efficiency quantification (Figure 8). sETR, irrespective of calculation method, also showed moderate correlations (r = 0.46 to 0.67) with growth traits, indicating the potential of using calculated sETR for crop growth prediction. Among the three HSI methods, the HSI linear method provided the lowest correlation, whereas the HSI quadratic and average methods showed relatively higher correlations. This demonstrates that the estimation of maximal fluorescence affects the capability of using sETR for growth prediction as well.

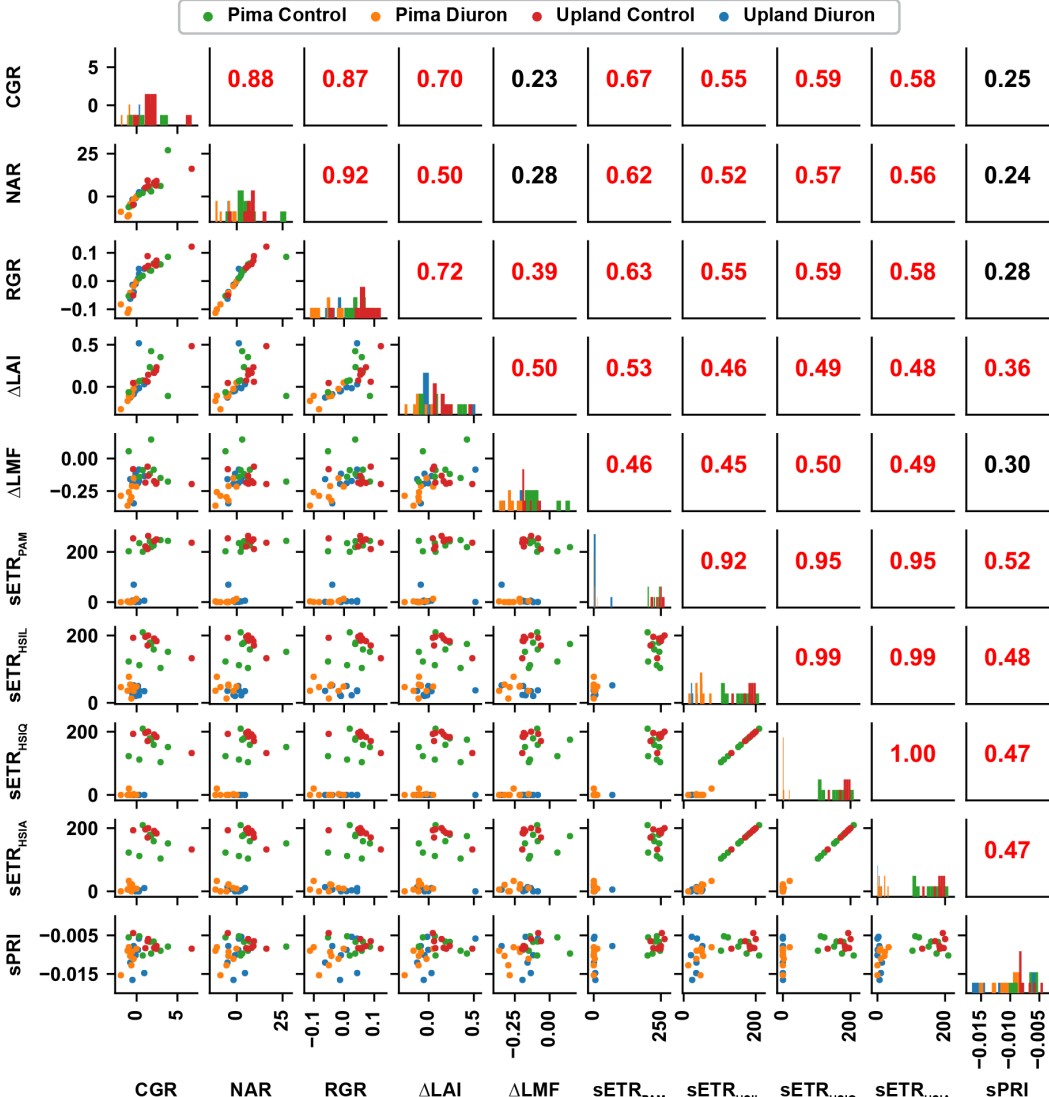

**Figure 8.** Pearson correlation values between growth traits and standardized ETR values estimated using four approaches. Values in the upper triangle are correlation values between traits indicated by a row and a column. Subplots along the main diagonal are the value distribution of individual traits, whereas subplots in the lower triangle are scattering plots between traits indicated by a row and a column. Significant (*p*-value < 0.05) correlation values are rendered by red color, and values are otherwise black color.

Compared with sETR, sPRI showed poorer correlations with other traits, especially for Diuron-treated plots. A possible reason was that PRI tends to reach a plateau value at a relatively low PAR. When the PAR value of 1500 $\mu$ mol/m$^2$/s was used, most sPRI fell into a small value range irrespective of plant growth status, resulting in a poor correlation with growth traits. This observation agreed with a previous study [59]. This suggests that the SIF-based characterization of photosynthetic activity might be broadly applicable under a range of environmental conditions.

## 4. Discussion

The developed approach used a passive hyperspectral imaging system to retrieve diurnal SIF values that were used for estimating maximal fluorescence, effective quantum yields of PSII, and ETR values at the canopy level. Irrespective of the approach utilized to model SIF to diurnal light intensity, the following conclusions can be made. Maximal fluorescence estimates for the canopy, derived from the best fit function of SIF versus the reciprocal of PAR in both the Diuron-treated plots and the control plots were used to calculate actual quantum yield of the canopy. $\phi_{HSI}$ estimates were strongly correlated with $\phi_{PAM}$ estimates when considered across multiple diurnal sampling times of all treatments or for the two different species of cotton evaluated (Figure 6). Furthermore, $\phi_{HSI}$ values were used to calculate ETR at a given diurnal PAR level, and diurnal light response curves were generated for both PAM and HSI-based methods. From these data, a standardized measure of ETR at a common light intensity (1500 µ mol/m$^2$/s) was calculated for both PAM and HSI-based methods. There was a strong correlation between HSI- and PAM-derived sETR values (r = 0.95; Figure 8), and canopy estimates of ETR were also predictive of whole-crop growth responses in the weeks following treatment. Thus, we suggest that is possible to estimate canopy-level photosynthetic efficiency from passive, diurnal measurements of SIF and PAR at the canopy at the time of measurement. This is a particularly notable achievement, since a number previous studies have collected diurnal SIF measurements at the agro-ecosystem scale in attempts to model gross primary productivity [12,60,61]. The methodology reported herein would allow researchers to obtain a direct measure of ecosystem-scale photosynthetic efficiencies from existing data sets. Furthermore, the use of hyperspectral imaging to delineate plot-scale photosynthetic efficiencies would allow for direct selection of genotypes with higher diurnal photosynthetic activities. Last, the developed method measures chlorophyll fluorescence in field conditions, which could expand fluorescence-based early detection of plant stresses (described by [62–64]) from the laboratory to the field. This could be particularly useful for agricultural production systems.

The developed approach has two major limiting factors, however. First, ETR values measured by HSI data showed a larger absolute differences than those measured by PAM data. This can be solved by improving the spectral sampling resolution [65]. The hyperspectral camera and spectrometer used the spectral sampling resolution of 2 nm in the present study, utilizing only around 50% depth of the Fraunhofer $O_2$-A line [65]. This limits the SIF retrieval accuracy, and thus, the derivation of effective quantum yield of PSII and ETR values. On the contrary, the spectral sampling resolution of the hyperspectral camera and downwelling spectrometer can be configured to approximately 0.2 nm, which could provide greater potential for further improvement of measurement accuracy. The finest spectral resolution will result in a large increase of data volume, presenting potential challenges in data collection, management, and processing. In addition to the hardware improvement, other SIF retrieval methods (e.g., 3FLD and improved FLD) can be used to increase the retrieval accuracy [24]. Second, estimation of maximal fluorescence dramatically affected the calculations of effective quantum yield, ETR, and RLC models. In the present study, a linear model provided the poorest results in which control and Diuron-treated groups showed a large overlap. Although fitted RLC models still showed a significant difference between treatments, it could be problematic to identify subtle differences of photosynthetic efficiency among genotypes. This could limit the potential of using the developed method for genetics/genomics studies and breeding programs. Future studies, therefore, need to validate the efficacy of the three models used in the present study and examine new ways to estimate maximal fluorescence using passive sensing methods. For instance, maximal fluorescence estimation models for stressed plants can be established and validated in a full PAR range from 0 to 6000 µ mol/m$^2$/s in a controlled environment, and transferred to field applications where data are acquired in a part of the full PAR range.

We should also acknowledge some other limitations identified in the present study. First, the present study was based on a single-year, single-location field experiment, which might present concerns (e.g., environment variations) from the agronomic viewpoint. The present study, however,

focused on methodology development to advance sensing and data analytics that can be potentially used for agricultural applications. Specifically, we aimed to develop and validate a ground hyperspectral imaging-based approach for diurnal SIF measurement and canopy-level photosynthesis characterization. In this context, diurnal data collection and trait extraction can be considered repeated experiments. High correlations between imaging- and fluorometer-derived measurements suggest high accuracy and repeatability of the developed system and analysis method, which, together, meet the study goal and objectives. Nonetheless, multi-year, multi-location experiments will be required to reveal agronomic and physiological findings in the future. Second, we only validated the developed approach through measurements (i.e., effective quantum yield of PSII and ETR) that can be obtained from a PAM-based fluorometer, which might limit the use of the approach for studies broadly related to photosynthesis. It is necessary to conduct successive experiments to comprehensively validate the developed approach through measurements that can be obtained by other well-established methods, such as SIF from instruments with high spectral resolution and gas exchange measurements.

## 5. Conclusions

The developed method showed promising results when using passive hyperspectral data to estimate effective quantum yield, ETR, and RLC models for the whole canopy. Patterns observed using calculated RLC characteristics agreed closely with growth traits, indicating that the developed method can be used to differentiate plants under extreme differences in photosynthetic efficiency. Regression analysis results confirmed that the calculated values had potential for plant growth prediction. Future studies will focus on exploring various estimation methods for maximal fluorescence of the canopy and potential applications in breeding programs and genetics/genomics studies.

**Author Contributions:** Conceptualization, Y.J., J.L.S., and C.L.; methodology, Y.J. and J.L.S.; software, Y.J.; validation, Y.J. and J.L.S.; formal analysis, Y.J.; investigation, Y.J. and J.L.S.; resources, J.L.S., C.L., G.C.R., and A.H.P.; data curation, C.L.; writing—original draft preparation, Y.J., J.L.S., and C.L.; writing—review and editing, Y.J., J.L.S., C.L., G.C.R., and A.H.P.; visualization, Y.J.; supervision, J.L.S. and C.L.; project administration, Y.J., J.L.S., and C.L.; funding acquisition, Y.J., C.L., and A.H.P. All authors have read and agreed to the published version of the manuscript.

**Funding:** This study was funded jointly by the Agricultural Sensing and Robotics Initiative of the College of Engineering, and the College of Agricultural and Environmental Sciences of the University of Georgia. The project was also partially supported by the National Robotics Initiative (NIFA grant number: 2017-67021-25928) and the University of Georgia Graduate School (Summer Research Travel Grant).

**Acknowledgments:** The authors gratefully thank Rui Xu, Gary Burnham, Ricky Fletcher, and personnel from Snider's and Rains' labs who contributed to this project with their assistance in planting, field management, data collection, and destructive growth analysis.

**Conflicts of Interest:** The authors declare no conflict of interest.

## Abbreviations

The following abbreviations are used in this manuscript:

| | |
|---|---|
| MDPI | Multidisciplinary Digital Publishing Institute |
| DOAJ | Directory of open access journals |
| TLA | Three letter acronym |
| LD | linear dichroism |

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
