# Peer review of "Ground Based Hyperspectral Imaging to Characterize Canopy-Level Photosynthetic Activities"

_remotesensing, doi:10.3390/rs12020315_

Round 1
Reviewer 1 Report
The manuscript has been improved after revision. I suppose that it can be accepted.
Author Response
We thank for the reviewer’s constructive suggestions that help to significantly improve the manuscript quality.
Reviewer 2 Report
The authors have addressed most of my questions. The manuscript is also improved. The topic of this study should be interested in the precision agriculture community. The experiment design and data analysis in the study are also well. However, as the authors said in the cover letter, there are so many uncertainties in field experiments. Thus, we usually request at least one repeat of the results for field experiments. Or the experiments should be conducted at several different sites. Therefore, I would suggest the authors repeat the experiment for at least once. Then the paper should be proper for publication.
Author Response
We thank the reviewer’s previous comments and suggestions that contribute to the considerable improvement of the manuscript quality. We understand the reviewer’s concern and the significance of two-year experiments for agronomic studies.
The present study focused on the methodology development and needs to be considered as an article that emphasizes the engineering advancement for agricultural applications. Specifically, we aimed to develop and validate a ground hyperspectral imaging-based approach for diurnal SIF measurement and canopy-level photosynthesis characterization. In this context, a single-year, single location experiment could be acceptable. In particular, the diurnal experiment included repeated data collection and SIF extraction, which can be a way to test the repeatability of the proposed sensing system and analysis method. Experiment results showed that hyperspectral-imaging derived SIF measurements are highly correlated with those measured by fluorometry, suggesting the good repeatability and accuracy of the system and analysis method. In addition, two publications used only a single-year, single location experiment to demonstrate the system and methodology development, suggesting that this practice is acceptable for the engineering advancement for agricultural applications (Pinto, F., et al. (2016). Sun‐induced chlorophyll fluorescence from high‐resolution imaging spectroscopy data to quantify spatio‐temporal patterns of photosynthetic function in crop canopies. Plant, cell & environment, 39(7), 1500-1512. and Rossini, M., et al. (2015). Red and far red Sun‐induced chlorophyll fluorescence as a measure of plant photosynthesis. Geophysical research letters, 42(6), 1632-1639.). We hope the reviewer could understand and accept this engineering practice.
To reflect this point, an additional paragraph was added in the Discussion.
“… We acknowledge that the present study was based on a single-year, single-location field experiment, which might present concerns (e.g., environment variations) from the agronomic viewpoint. The present study, however, focused on methodology development to advance sensing and data analytics that can be potentially used for agricultural applications. Specifically, we aimed to develop and validate a ground hyperspectral imaging-based approach for diurnal SIF measurement and canopy-level photosynthesis characterization. In this context, diurnal data collection and trait extraction can be considered as repeated experiments. High correlations between imaging- and fluorometer-derived measurements suggest high accuracy and repeatability of the developed system and analysis method, which meets with the study goal and objectives. Nonetheless, multi-year, multi-location experiments will be required to reveal agronomic and physiological findings in the future. …”
Round 2
Reviewer 2 Report
Even there have been several articles with single site and single year experiments published. It does not mean that the common agricultural community will accept that.
Anyway, as I mentioned before, there are few problems in this article now except for the repeat of the experiment.
In general, I am not opposed to the publication of this article. Thus, I would leave the decision for the academic editors of Remote Sensing.
Author Response
Response to Comments from Reviewer #2
C1: Even there have been several articles with single site and single year experiments published. It does not mean that the common agricultural community will accept that. Anyway, as I mentioned before, there are few problems in this article now except for the repeat of the experiment. In general, I am not opposed to the publication of this article. Thus, I would leave the decision for the academic editors of Remote Sensing.
Response:
We thank the reviewer for providing many good suggestions to improve the manuscript quality during the review process. We revised the manuscript based on the editor’s suggestions and hope they can further improve the manuscript. To reflect this, we listed our responses to the editor’s comments as follow.
Response to Editor’s Comments
C1: We found that most of ground-based SIF measurements are based on the spectroradiometer of 0.1-0.2 nm, however, the spectral imagine of your case has 2nm spectral resolution. So it is necessary and important to evaluate your derived SIF with either the gold-standard (i.e. ground SIF measurements of 0.1-0.2 nm) or direct physiological measurements (i.e. gas exchange measurements through LiCOR). We found this part is yet lacking, and would appreciate if you can either provide some more direct evidence to support your work or well acknowledge the validation limitation of your current work.
Response:
Thanks for the editor’s suggestion. We agree that the current work has a limitation in the reference measurements (neither the gold-standard high-resolution of SIF measurement nor gas exchange measurement). In particular, gas exchange measurement is a direct approach and will provide net photosynthesis, which is the most important parameter. We chose ETR and \phi_{PSII} obtained from PAM-based fluorometer for two considerations. First, we try to measure reference values using a technique that is well-established but based on a principle different than SIF, which can better validate the SIF technique. Second, compared with gas exchange measurements (and thus net photosynthesis which can be influenced by respiration), ETR is more directly applicable to SIF. Nonetheless, we followed the editor’s suggestion and well acknowledged the validation limitation in the current work.
“… We also acknowledge some other limitations identified in the present study. First, the present study was based on a single-year, single-location field experiment, which might present concerns (e.g., environment variations) from the agronomic viewpoint. The present study, however, focused on methodology development to advance sensing and data analytics that can be potentially used for agricultural applications. Specifically, we aimed to develop and validate a ground hyperspectral imaging-based approach for diurnal SIF measurement and canopy-level photosynthesis characterization. In this context, diurnal data collection and trait extraction can be considered as repeated experiments. High correlations between imaging- and fluorometer-derived measurements suggest high accuracy and repeatability of the developed system and analysis method, which meets with the study goal and objectives. Nonetheless, multi-year, multi-location experiments will be required to reveal agronomic and physiological findings in the future. Second, we only validated the developed approach through measurements (i.e., effective quantum yield of PSII and ETR) that can be obtained from a PAM-based fluorometer, which might limit the use of the approach for studies broadly related to photosynthesis. It is necessary to conduct successive experiments to comprehensively validate the developed approach through measurements that can be obtained by other well-established methods, such as SIF from instruments with high spectral resolution and gas exchange measurements. …”
C2: Please carefully check on the regression statistics of your Fig. 6, which might be wrongly spelled, especially for Fig. 6 b,c,e,f.
Response:
Thanks for the editor’s checking. We have corrected the duplicated symbols in the regression results in Figure 6 a and d. We also double-checked the statistical results presented in Figure 6 b, c, e, and f. Correlations considerably increased because the quadratic and average models provided better estimate of maximal fluorescence and thus quantum yield measurements.
This manuscript is a resubmission of an earlier submission. The following is a list of the peer review reports and author responses from that submission.
Round 1
Reviewer 1 Report
The manuscripts investigated the potential of using a ground-based imaging sensor to study canopy level photosynthetic activity. While the paper is well written and comprehensive, there are some points that need to be addressed before it can be published.
Line 15: Please define what is DCMU Line 43: Please cite Line 46: Instead of saying [9], however, suggested… replace with Constable [9], however, suggested, … Line 63: Maybe start a new sentence with “There are also several other derived variables...” Line 73: remove the first “and” Line 75-76: Suggest changing to “…illumination, which limits…” In the introduction, you mention some systems being used to measure SIR, but I would suggest expanding that discussion on other ground-based HSI systems that have been utilized recently to study different biophysical characteristics. I suggest a more elaborate discussion on the use of ground-based HSI system in retrieving different biophysical properties of vegetation, e.g. chlorophyll, aboveground biomass, LAI, etc… Line 102-104: Please cite Line 158: Please mention how the two different systems were calibrated Line 194: Please elaborate on why you use those two wavelength and band ratio to create a mask for the plants What was the GSD from the imaging system? I would like to see a RGB image of the cotton plants taken by the HSI system, maybe in Figure 1, to give readers a better idea on its spatial resolution. The PAR curve in Figure 1, is it an average of all pixels within the column or just a single pixel for each column. Please specify in the manuscript. If you added more data points, would you expected to see an exponential relation for the two Diuron combinations? It is a little hard to accept those results since the authors did not see any correlation between SIF and PAR. It is also unfortunate that the authors were not consistent in taking the maximum SIF values and had to use the mean for some instances. Line 237: Please cite where you get the C3 value from Line 315: Missing bracket Line 328: Why was it reasonable to use the average model Line 330: what could be some of the measurement errors I am a little confused with sentence in line 390, since from figure 6 I see no correlation at all for the Diuron data A lot is going on in Figure 8, and would like to see more of a discussion on it in section 3.5 Line 437: I don’t think you can make that statement, since the diuron data really didn’t show any correlation in figure 6, only correlation is seen with the control dataAuthor Response
Response to Comments from Reviewer #1
Q1 Line 15: Please define what is DCMU
Response:
The full chemical term was added to the first place where DCMU used.
Q2 Line 43: Please cite
Response:
A reference (Hedden, P. (2003). The genes of the Green Revolution. TRENDS in Genetics, 19(1), 5-9.) was cited.
Q3 Line 46: Instead of saying [9], however, suggested… replace with Constable [9], however, suggested, …
Response:
Thanks for the reviewer’s careful check. We added “A study” before citation [10] in the sentence.
Q4: Line 63: Maybe start a new sentence with “There are also several other derived variables...”
Response:
Revised as suggested.
Q5: Line 73: remove the first “and” Line 75-76: Suggest changing to “…illumination, which limits…”
Response:
Revised as suggested.
Q6: In the introduction, you mention some systems being used to measure SIR, but I would suggest expanding that discussion on other ground-based HSI systems that have been utilized recently to study different biophysical characteristics. I suggest a more elaborate discussion on the use of ground-based HSI system in retrieving different biophysical properties of vegetation, e.g. chlorophyll, aboveground biomass, LAI, etc…
Response:
Based on the reviewer’s comment, we added the following sentences to discuss previous efforts on using hyperspectral imaging for estimation of aboveground biomass, leaf area index, and plant pigments.
“Ground-based hyperspectral imaging would be a viable solution to address those issues because it provides a higher scanning throughput (usually line by line) and a better spatial resolution (sub meter or higher). Many researchers have explored the use of ground-based hyperspectral imaging for extracting vegetative indices that can be correlated with plant aboveground biomass [34], leaf area index [35], and various plant pigments (e.g., chlorophyll) [36]. Although these studies showed certain success, they required either sophisticated model calibration and validation [36] or artificial illumination [34].”
[34]: Yao, X., et al., Hyperspectral estimation of canopy leaf biomass phenotype per ground area using a continuous wavelet analysis in wheat. Frontiers in plant science, 2018. 9.
[35]: Din, M., et al., Evaluating hyperspectral vegetation indices for leaf area index estimation of Oryza sativa L. at diverse phenological stages. Frontiers in Plant Science, 2017. 8: p. 820.
[36]: Blackburn, G.A., Hyperspectral remote sensing of plant pigments. Journal of experimental botany, 2006. 58(4): p. 855-867.
Q7: Line 102-104: Please cite
Response:
While there are many well-respected and heavily cited papers that have said the same thing, we have added the following reference because it clearly dives into the biophysical basis of PAM measurements and compares them with the information that may be obtained from SIF measurements. Specifically, in the below paper the authors state the following:
“Observations of SIF will only see the steady state level. Information on is beyond the reach of the method.”
Van der Tol, C., et al. "Models of fluorescence and photosynthesis for interpreting measurements of solar‐induced chlorophyll fluorescence." Journal of Geophysical Research: Biogeosciences, 119.12 (2014): 2312-2327.
We cited this paper for reference.
Q8: Line 158: Please mention how the two different systems were calibrated
Response:
Description of system calibration method was added as follow.
“For the spectral calibration, the spectrometer was calibrated by the manufacturer, whereas the hyperspectral camera was calibrated by the authors using three calibration lamps. Details of spectral and spatial calibration can be found in a previous study [37]. For the radiometric calibration, the spectrometer, the hyperspectral camera, a spectroradiometer (Ocean Optics Inc., Largo, FL, USA), and an illumination source (DC-950, Fiber-Lite, Dolan-Jenner Industries, Boxborough, MA) were attached to an integrating sphere (4P-GPS-060-SF, Labsphere, North Sutton, NH). By adjusting the illumination intensity at different levels, calibration models between digital counts and irradiance were established for the spectrometer and hyperspectral camera.”
Q9: Line 194: Please elaborate on why you use those two wavelength and band ratio to create a mask for the plants What was the GSD from the imaging system? I would like to see a RGB image of the cotton plants taken by the HSI system, maybe in Figure 1, to give readers a better idea on its spatial resolution.
Response:
Based on preliminary tests, the two wavelengths were used to calculate the ratio for creating plant masks. Text was revised as follow.
“Based on preliminary tests, grayscale images at 749 nm and 685 nm were used to generate a band ratio image.”
The GSD of the imaging system was 1.7 mm/pixel. Below is an RGB image produced using 475 nm (blue), 545 nm (green), and 651 nm (red), respectively, for a part of a plot in the field. To better demonstrate the spatial resolution, we will upload the raw RGB image generated by the three wavelengths as a supplementary material.
Q10: The PAR curve in Figure 1, is it an average of all pixels within the column or just a single pixel for each column. Please specify in the manuscript.
Response:
If we understand correctly, the PAR curve refers to the one in Figure 2. A sentence was added to explain it’s an average of all pixels within the column.
“One PAR value corresponded to the average value for all pixels in one line of a hyperspectral cube.”
Q11: If you added more data points, would you expected to see an exponential relation for the two Diuron combinations? It is a little hard to accept those results since the authors did not see any correlation between SIF and PAR. It is also unfortunate that the authors were not consistent in taking the maximum SIF values and had to use the mean for some instances.
Response:
Based on the current results, if only more data points were added, it might not be likely to have an exponential relationship for the two Diuron treatments. We understand the reviewer’s concern. This is a challenge of using SIF to characterize canopy-level photosynthetic activities. We hope the present models would be a starting point that can inspire future studies.
Q12: Line 237: Please cite where you get the C3 value from
Response:
Two references (Yin, X. and P. Struik, C3 and C4 photosynthesis models: an overview from the perspective of crop modelling. NJAS-Wageningen Journal of Life Sciences, 2009. 57(1): p. 27-38.) and (Flexas J, Escalona JM, Medrano H (1998) Down-regulation of photosynthesis by drought under field conditions in grapevine leaves. Australian Journal of Plant Physiology 25, 893–900.) were cited for this.
Q13: Line 315: Missing bracket
Response:
Thanks for the reviewer’s careful check. A bracket was added.
Q14: Line 328: Why was it reasonable to use the average model
Response:
If the measurement variation is considered as measurement noise, the average model would be a reasonable option to reduce the noise level.
Q15: Line 330: what could be some of the measurement errors I am a little confused with sentence in line 390, since from figure 6 I see no correlation at all for the Diuron data A lot is going on in Figure 8, and would like to see more of a discussion on it in section 3.5
Response:
We understand why the wording used in this section of the manuscript confused the reviewer, given the fact that there does not appear to be a strong correlation between quantum yield values obtained using either method for the diuron group. However, we are discussing different models for estimating Fm’ at infinite light intensity, and the important thing to note is when the “best fit” function is used to estimate Fm’ (regardless of herbicide treatment), we have the strongest relationship between HSI and PAM based quantum yield data. This is when all quantum yield data are included from both herbicide treatments. Regarding the apparent lack of a relationship between quantum yields for the diuron only group, irrespective of method used (PAM or HSI), the values for Diuron treated plots were inherently low (between 0 and up to values less than 0.5) because our treatment was applied to abolish electron transport to inter-system electron acceptors. Thus, the range of data was inherently small if we only look at one treatment, and no apparent relationship exists, but our intent was to include the diuron treatment to provide values near zero, which we couldn’t do with the control plots, and include the healthy plots to provide our higher values that we couldn’t get with our diuron plots. This ultimately gives us a broader range of data over which to test our method, and what figure 6 tells us is the following. Over the entire range of quantum yield data obtained in this study, when we use the best fit models (quadratic for the diuron groups and exponential for the control groups; see Figure 4) to estimate Fm’ at infinite light intensity, we obtain strong relationships between HSI and PAM methods (r2 = 0.895 for Pima cotton and 0.935 for Upland cotton; Fig. 6b,e)
Q16: Line 437: I don’t think you can make that statement, since the diuron data really didn’t show any correlation in figure 6, only correlation is seen with the control data
Response:
The statement was revised to highlight the overall correlation rather the correlation for Diuron treatments.
“φHSI estimates were strongly correlated with φPAM estimates when considered across multiple diurnal sample times of all treatments or for the two different species of cotton evaluated”

Reviewer 2 Report
The manuscript by Jiang et al. is devoted interesting problem: development of method of estimation of the photosystem II quantum yield on basis of analysis of the sun-induced fluorescence. However, there are the number of questions and comments.
Title
I am not fully sure that term “Hyperspectral Imaging” is correct in the Title, because the work is devoted to analysis of the sun-induced fluorescence.
Abstract
P. 1, line 8: I am not fully sure that term “Hyperspectral Imaging” is correct in this sentence.
P. 1, line 17: Time format “0800” and “1800” does not seem to be correct; “08:00” and “18:00” are rather correct. This format (without “:”) was also used in other parts of the manuscript.
Introduction
Often, hyper- and multispectral imaging is based on calculation of reflectance indices. In particular, the photochemical reflectance index (PRI) can be used for monitoring of fast changes in photosynthetic activity. There are numerous works (including several works with meta-analysis of literature data), which analyze connection of PRI with the photosystem II quantum yield, the non-photochemical quenching, etc. It is very probable that analysis of PRI (in particular, with using of hyperspectral camera) can be effective alternative of analysis of SIF in remote sensing of photosynthetic processes. As a result, I suppose that using of PRI for the photosynthetic remote sensing should be described in details, both methods (measurements of SIF and PRI) should be compared and selection of SIF for investigation should be proved.
I suppose that alternative models connecting SIF and photosynthetic parameters (which is not based on “PAM-like” method) should be described.
P. 2, lines 60-68: Difference between Fo and Fo’. Fm and Fm’, etc. should be clarified in the text.
P.2 , lines 72-74: Fo, Fm, Fs, etc. were described above.
Materials and methods
P. 4, lines 157-156: What was spectral resolution of spectrometer and, especially. hyperspectral camera (MSV500)?
Figure 1a shows that the used mobile platform can shade zone of measurement by the hyperspectral camera; however, this effect must be absent of spectrometer. Can it influence results?
P. 5, lines 171-182: Fluorescence measurements by PAM method should be described in more details. What measuring light, saturation light and actinic light were used? What were intensity and spectral maximum? What duration of dark adaptation was used? etc.
P. 6, lines 206-209: Why Fraunhofer line at 760 nm was used?
P. 6, lines 206-209: Why 759 and 761 nm were used for analysis? In accordance with Figure 2, width of the line seems to be about 5-10 nm. For example, why 755 and 761 was not used? Increase of spectral distance between analyzed bands can simplify measurements. Additionally, two neighboring bands (e.g. 755 and 767 nm) can be measured and averaged. It should be clarified.
P. 7, lines 210-211, Equation (1): It is not clear. E was measured be spectrometer; however, spectrometer does not measure light in single pixels. How E in pixel was measured?
P. 7, Equation (3): SIFm was calculated as averaged value for all plants in the experimental variant or as the individual value? It should be clarified.
P. 7, Equation (4), line 237: If PAR was calculated as total flow of photons at 400-700 nm (Equation 2) then using of Equation (4) is not simple. In particular, the leaf absorbance of incident light is dependent on light wavelength (Figure 2 supports it). This question should be clarified.
P. 8, lines 246-262: I suppose that necessity of application of rapid light curves in the work should be described in more details; it is possible that this explanation will simplify understanding of work by readers.
P. 8, Equation (6): This equation should be clarified, In particular, if a>0 (in accordance with Table 1) then ETR must be changed from 0 – to -∞. I suppose that “-“ is necessary (in this case, ETR will be changed from 0 – to mETR). Additionally, mETR in exp(a*PAR/mETR) is not understand; it should be clarified.
P. 8, Equation (8): It is not clear – how the equation is described the net assimilation rate (NAR); it should be clarified.
Results
Figures 3 and 4: Why diuron suppressed intensity of SIF? The diuron inhibited electron transfer from Qa – to Qb; as a result number of “closed” reaction centers of PSII (centers with Qa-) should be maximal after treatment by diuron and fluorescence should be maximal. This problem should be clarified.
P. 9, lines 285-287: “In contrast, SIF values for diuron-treated plots exhibited no relation with PAR values: SIF values were low and relatively constant irrespective of PAR changes throughout the day”. However, light dependence of SIF after treatment by diuron can be approximated by functions of light intensity. This contradiction should be clarified.
Figures 5 and 6: The figures show that the quantum yield of PSII, which was calculated on basis of PAM-method, and this yield, which was calculated on basis of SIF, were weakly connected after treatment by diuron. Does it mean that SIF can not be used for calculation of the quantum yield of PSII at these conditions?
Figure 6: Was connection between the quantum yields of PSII, estimated on basis of PAM-method and on basis of SIF, analyzed in control only (without variants, which was treated by diuron)? It can be interesting because connection between these quantum yields after treatment by diuron seems to be absent.
Discussion
P. 15, line 438: Sentence “… in response to Diuron treatment …” is not fully correctness because connection between quantum yields of PSII, which was calculated on basis of PAM-method and SIF, after treatment by diuron seems to be absent.
P. 17, lines 461-463: It is not clear – why spectral resolution about 0.2 nm can improve method? It should be clarified. I think that the method can be rather improved by increase of spectral distance between investigated spectral bands and/or by using two neighboring spectral bands (see above).
P. 17-18, lines 465-475: Using of preliminary estimated SIFm (e.g. at action of different stressors) can be not effective for investigation of group of plants with different values of stressor-induced changes in photosynthetic processes. It is very probable that action of stressor in field conditions will induce different values of stress responses in different plants.
Additional general point: text of the manuscript seems to be chaotic; there are small errors (e.g. P. 18, lines 486-492: Author Contributions was not completed). I suppose that additional check of the manuscript can improve it.
Thus, I suppose that revision is necessary.
Author Response
Responses to Comments from Reviewer #2
Q1: I am not fully sure that term “Hyperspectral Imaging” is correct in the Title, because the work is devoted to analysis of the sun-induced fluorescence.
Response:
While a core part of this study is to extract solar induced fluorescence (SIF), hyperspectral imaging is the key technology used in the study for SIF retrieval, which is dramatically different from most existing literature. Thus, we think the term “hyperspectral imaging” is necessary to be mentioned in the title.
Q2: P. 1, line 8: I am not fully sure that term “Hyperspectral Imaging” is correct in this sentence.
Response:
This is the same comment to Q1 and please refer to our Q1’s responses
Q3: P. 1, line 17: Time format “0800” and “1800” does not seem to be correct; “08:00” and “18:00” are rather correct. This format (without “:”) was also used in other parts of the manuscript.
Response:
Based on the reviewer’s comment, the time format was revised to “hh:ss” throughout the manuscript.
Q4: Often, hyper- and multispectral imaging is based on calculation of reflectance indices. In particular, the photochemical reflectance index (PRI) can be used for monitoring of fast changes in photosynthetic activity. There are numerous works (including several works with meta-analysis of literature data), which analyze connection of PRI with the photosystem II quantum yield, the non-photochemical quenching, etc. It is very probable that analysis of PRI (in particular, with using of hyperspectral camera) can be effective alternative of analysis of SIF in remote sensing of photosynthetic processes. As a result, I suppose that using of PRI for the photosynthetic remote sensing should be described in details, both methods (measurements of SIF and PRI) should be compared and selection of SIF for investigation should be proved.
Response:
Following the reviewer’s suggestion, we added details of PRI and the rationale of choosing SIF over PRI in the present study.
“It should be noted that passive hyperspectral sensing systems can also measure canopy reflectance indices to monitor photosynthetic activities. For instance, photochemical reflectance index (PRI) was introduced by Gamon et al. [39], and has been studied extensively [40-42]. It is calculated as (R531-R570)/(R531+R570) and has been correlated with photosynthetic efficiencies previously. However, some studies reported that PRI is somewhat limited in that measurements are highly sensitive to viewing angle and are strongly influenced by soil background at leaf area index (LAI) less than 3 [12,41]. In particular, a previous study has found no correlation between PRI and light use efficiency (LUE) [43]. Thus, it would be of great interest to focus on measurement of SIF rather other reflectance indices for photosynthetic activity evaluation, especially at the canopy level.”
In addition, PRI-based analysis was conducted, so the potential of using PRI and SIF for characterization of photosynthetic activity was evaluated and compared. Details of analysis method and results were in the section 2.4 and 3.5. Figure 8 was accordingly updated with the PRI analysis results.
Q5: I suppose that alternative models connecting SIF and photosynthetic parameters (which is not based on “PAM-like” method) should be described.
Response:
Following the reviewer’s suggestion, descriptions/references are added.
“… Multiple studies have shown strong correlations between gross primary productivity (whole canopy photosynthesis) and SIF through various modeling methods [12, 28, 29]. When using SIF tracks GPP, there is usually a strong association with the absorbed photosynthetically active radiation (PAR). In situations where light intensity is constant and extreme stress (such as heat stress) limits canopy photosynthesis, however, the relationship is somewhat degraded because excess energy that cannot be dissipated through non-photochemical quenching (NPQ), might be emitted as increased fluorescence [30]. Thus, it is important to develop a method to generate canopy level quantum yields and photosynthetic activities if the method can be broadly applicable under a range of environmental conditions. …”
Q6: P. 2, lines 60-68: Difference between Fo and Fo’. Fm and Fm’, etc. should be clarified in the text.
Response:
Variables with or without the prime are for fluorescence state measurements from dark- or light-adapted leaf samples. This difference was explained in the same paragraph: “Variables denoted by prime are for light-adapted states, or otherwise for dark-adapted states.”
Q7: P.2 , lines 72-74: Fo, Fm, Fs, etc. were described above.
Response:
Please refer to responses to Q6.
Q8: P. 4, lines 157-156: What was spectral resolution of spectrometer and, especially. hyperspectral camera (MSV500)?
Response:
The spectrometer has an average spectral resolution of 1.5 nm (FWHM) and spectral sampling interval of 0.5 nm. The hyperspectral camera has a nominal spectral resolution of 2.8 nm and spectral sampling interval of 2 nm.
Q9: Figure 1a shows that the used mobile platform can shade zone of measurement by the hyperspectral camera; however, this effect must be absent of spectrometer. Can it influence results?
Response:
Yes, shading will dramatically affect results of SIF extraction and therefore analysis of photosynthetic activities. In this study, experimental plots was planted along the east-west direction, so the system moved sunward to avoid shades on the canopy surface being scanned. Figure 1a only illustrates the system rather the actual scanning in the field. To clarify this, additional descriptions were added.
“… Plots were arranged along the east-west direction. …”
“… The system moved sunward along the plot direction (east-west direction) to avoid potential issues caused by shading effects. ...”
Q10: P. 5, lines 171-182: Fluorescence measurements by PAM method should be described in more details. What measuring light, saturation light and actinic light were used? What were intensity and spectral maximum? What duration of dark adaptation was used? etc.
Response:
Firstly, it is general accepted that a true estimate of Fm’ is not achievable for plants grown under high light intensity, so the multi-phase flash protocol is used to estimate Fm’ when all reaction centers are closed. This was already described and light intensities at each phase of the multi-phase flash were provided. We have added text specifying that each light intensity from the multi-phase flash was provided by a 35 W halogen bulb. We have also added some additional text to very briefly describe the measuring (modulation) light source, and make it clear to readers that naturally-occurring solar irradiance was our actinic light source.
“At each diurnal sampling time, the leaf blade was clipped so that the orientation of the exposed adaxial surface relative to incoming solar radiation was left unchanged, and steady state fluorescence (Fs’) was measured under ambient light conditions using a 660 nm modulation measuring beam under naturally occurring solar irradiance as our actinic light source. While measuring fluorescence, PAR at the leaf surface was estimated using a PAR sensor integrated into the leaf clip. Subsequently, maximal fluorescence intensity (Fm’) was estimated using a multi-phase flash (provided by a 35Whalogen bulb) approach comparable to the methods described in [39], where relative fluorescence intensity is plotted versus the reciprocal of PAR following exposure of the leaf sample to a sequence of flashes with increasing intensity (2850, 5700, and 8550 mmol/m2/s) for a total duration of 0.95 s.”
Regarding dark adaptation, this would only be relevant if we were measuring maximum quantum yield (Fv/Fm), which we were not. We were measuring, actual quantum yield under a given, naturally occurring, actinic light intensity (ΦPSII). Equation 5 clearly defines its calculation, and dark adaptation is never used for doing these measurements (Maxwell and Johnson, 2000).
Q11: P. 6, lines 206-209: Why 759 and 761 nm were used for analysis? In accordance with Figure 2, width of the line seems to be about 5-10 nm. For example, why 755 and 761 was not used? Increase of spectral distance between analyzed bands can simplify measurements. Additionally, two neighboring bands (e.g. 755 and 767 nm) can be measured and averaged. It should be clarified.
Response:
Although three Fraunhofer lines (Hα ~656nm, O2-B ~687 nm, and O2-A ~760 nm) can be potentially used for SIF retrieval, O2-A (~760 nm) was used for two reasons. First, it can provide the best signal depth (thus extraction accuracy) under a lower spectral resolution. Second, it is closer to the actual chlorophyll fluorescence emission at approximately 740 nm. (Meroni, M., et al., Remote sensing of solar-induced chlorophyll fluorescence: Review of methods and applications. Remote Sensing of Environment, 2009. 113(10): p. 2037-2051.). The same reference suggests 758 nm as a common out-of-line wavelength for SIF retrieval. In the present study, 759 nm and 761 nm were used for SIF retrieval because they are the wavelengths closest to the suggested ones.
We understand the reviewer’s concern that advanced SIF retrieval models can be used, but the present study aims to explore the possibility of using a ground hyperspectral imaging system for SIF retrieval and successive canopy-level photosynthetic analysis. Thus, it is preferred to use conventional parameters to establish a baseline in this study.
To reflect these points, the text was revised as follow.
“… In the present study, based on previous literature review [24], the O2-A band (approximately at 761 nm) and its neighboring band (759 nm) were used to calculate SIF values of individual pixels using Equation 1. It is noteworthy that the sFLD method was used because the present study focused on exploring the possibility of using hyperspectral imaging for SIF retrieval and successive canopy-level photosynthetic analysis. Other Fraunhofer line discrimination models (e.g., 3FLD and improved FLD) can be used to improve SIF retrieval accuracy in future studies [24]. …”
Q12: P. 7, lines 210-211, Equation (1): It is not clear. E was measured be spectrometer; however, spectrometer does not measure light in single pixels. How E in pixel was measured?
Response:
In the present study, the hyperspectral camera is line-scan based. We assumed that incident light was same for individual pixels in one line image.
“… Pixels along individual lines in a radiance hyperspectral cube shared the same irradiance spectrum measured by the spectrometer. ….”
Q13: P. 7, Equation (3): SIFm was calculated as averaged value for all plants in the experimental variant or as the individual value? It should be clarified.
Response:
SIFm was estimated for all plants in different combinations of treatment and cultivar. This was clarified in the text.
“… and the model y-interception (the reciprocal of PAR equaled to zero) was treated as the maximum SIF value for all plants in that combination. …”
Q14: P. 7, Equation (4), line 237: If PAR was calculated as total flow of photons at 400-700 nm (Equation 2) then using of Equation (4) is not simple. In particular, the leaf absorbance of incident light is dependent on light wavelength (Figure 2 supports it). This question should be clarified.
Response:
We understand the reviewer’s concern that wavelength affects light absorption by chlorophyll, which is well-established in plant physiology textbooks and primary literature. It does not change the fact that the equation given is valid for the purposes of our research. The absorption coefficient provided is a widely applied absorption coefficient for C3 species (0.84). It just means that of all the incoming PAR (all light in the wavelength range you noted previously), the leaf will absorb 84 percent of it. Of that light, a certain portion of it will be used to drive photochemistry, and we can estimate this efficiency (quantum yield) using fluorescence. This is a well-established equation in the literature used to estimate electron transport from quantum yield and light intensity at the leaf surface, but we did not make this clear in the previous version of the paper. Thus, we have added references for this in the revised version of the paper (Flexas et al. 1998; Yin et al. 2009) in response to your comment and the one by reviewer 1. This is actually so well-established in the scientific literature that this absorption coefficient is automatically built into calculations used by the LI-COR portable photosynthesis system (LI-6400 and LI-6800) to estimate ETR from quantum yield and light intensity. These instruments are the industry standard in photosynthesis systems and have been cited in tens of thousands of original research articles on photosynthesis. We feel confident in the equation provided.
Q15: P. 8, lines 246-262: I suppose that necessity of application of rapid light curves in the work should be described in more details; it is possible that this explanation will simplify understanding of work by readers.
Response:
We added the following text to the revised version of the manuscript.
“Because light intensity influences electron transport rates and can fluctuate throughout the day or even within the same measurement period for a given plot, rapid light response curves were generated from diurnal data to provide a standardized measure of maximum electron transport rate for each plot.”
Q16: P. 8, Equation (6): This equation should be clarified, In particular, if a>0 (in accordance with Table 1) then ETR must be changed from 0 – to -∞. I suppose that “-“ is necessary (in this case, ETR will be changed from 0 – to mETR). Additionally, mETR in exp(a*PAR/mETR) is not understand; it should be clarified.
Response:
Thanks for the reviewer’s careful check. There is a typo missing the “-” sign for the term . The equation was corrected.
“… (6)…”
Q17: P. 8, Equation (8): It is not clear – how the equation is described the net assimilation rate (NAR); it should be clarified.
Response:
In theory (also intuitively), the net assimilation rate is the rate of increase of dry weight (W) per unit of leaf area (A), namely, . In practice, dry weight is measured destructively, so measurements on two sampling days are from two batches of plant samples, which involves certain sampling errors. To correct this, an equation, , has been proposed and widely used for NAR measurement. (Vernon, A. J., & Allison, J. C. S. (1963). A method of calculating net assimilation rate. Nature, 200(4908), 814-814.).
Q18: Figures 3 and 4: Why diuron suppressed intensity of SIF? The diuron inhibited electron transfer from Qa – to Qb; as a result number of “closed” reaction centers of PSII (centers with Qa-) should be maximal after treatment by diuron and fluorescence should be maximal. This problem should be clarified.
Response:
This is a good question. Previous studies reported that Diuron inhibits electron transfer and thus maximized chlorophyll fluorescence signals right after chemical spray (Rossini, Md, et al., 2015 and Pinto, Francisco, et al., 2016). However, in the present study, we saw an opposite results likely due to the fact that Diuron was applied to cotton for a full 24-hour cycle before data collection and induced actual stresses for the plants. SIF measurements were from stressed plants rather plant being inhibited simply by electron transfer. Thus, a different pattern was observed. This was explained in the text.
“The control groups were healthy, showing fluorescence intensity changes along with varied PAR levels, whereas an inhibitor of electron transport beyond PSII would be expected to cause damage to the photosynthetic apparatus and potentially increase non-photochemical quenching (NPQ) of the fluorescence signal.”
Q19: P. 9, lines 285-287: “In contrast, SIF values for diuron-treated plots exhibited no relation with PAR values: SIF values were low and relatively constant irrespective of PAR changes throughout the day”. However, light dependence of SIF after treatment by diuron can be approximated by functions of light intensity. This contradiction should be clarified.
Response:
In the present study, plants treated with Diuron were induced to actual stress status. Thus, a different pattern was observed than that in previous studies where SIF measurements were conducted right after the application of Diuron. It should be noted that our observation was similar to studies that used SIF for plant stress detection.
Q20: Figures 5 and 6: The figures show that the quantum yield of PSII, which was calculated on basis of PAM-method, and this yield, which was calculated on basis of SIF, were weakly connected after treatment by diuron. Does it mean that SIF can not be used for calculation of the quantum yield of PSII at these conditions?
Response:
We understand why the wording used in this section of the manuscript confused the reviewer, given the fact that there does not appear to be a strong correlation between quantum yield values obtained using either method for the diuron group. However, we are discussing different models for estimating Fm’ at infinite light intensity, and the important thing to note is when the “best fit” function is used to estimate Fm’ (regardless of herbicide treatment), we have the strongest relationship between HSI and PAM based quantum yield data. This is when all quantum yield data are included from both herbicide treatments. Regarding the apparent lack of a relationship between quantum yields for the diuron only group, irrespective of method used (PAM or HSI), the values for Diuron treated plots were inherently low (between 0 and up to values less than 0.5) because our treatment was applied to abolish electron transport to inter-system electron acceptors. Thus, the range of data was inherently small if we only look at one treatment, and no apparent relationship exists, but our intent was to include the diuron treatment to provide values near zero, which we couldn’t do with the control plots, and include the healthy plots to provide our higher values that we couldn’t get with our diuron plots. This ultimately gives us a broader range of data over which to test our method, and what figure 6 tells us is the following. Over the entire range of quantum yield data obtained in this study, when we use the best fit models (quadratic for the diuron groups and exponential for the control groups; see Figure 4) to estimate Fm’ at infinite light intensity, we obtain strong relationships between HSI and PAM methods (r2 = 0.895 for Pima cotton and 0.935 for Upland cotton; Fig. 6b,e).
Q21: Figure 6: Was connection between the quantum yields of PSII, estimated on basis of PAM-method and on basis of SIF, analyzed in control only (without variants, which was treated by diuron)? It can be interesting because connection between these quantum yields after treatment by diuron seems to be absent.
Response:
Figure 6 was based on results of analyzing both treatments (control and Diuron). This is a comment related to Q20 and please see our response for Q20.
Q22: P. 15, line 438: Sentence “… in response to Diuron treatment …” is not fully correctness because connection between quantum yields of PSII, which was calculated on basis of PAM-method and SIF, after treatment by diuron seems to be absent.
Response:
Thanks for the reviewer’s comment. We corrected the sentence to highlight the overall correlation rather the correlation in Diuron treatments.
“… when considered across multiple diurnal sample times of all treatments or for the two different species of cotton evaluated …”
Q23: P. 17, lines 461-463: It is not clear – why spectral resolution about 0.2 nm can improve method? It should be clarified. I think that the method can be rather improved by increase of spectral distance between investigated spectral bands and/or by using two neighboring spectral bands (see above).
Response:
This is a good point and we understand this. On the other hand, if the hardware can have a better spectral resolution (therefore a deeper signal depth), SIF retrieval accuracy could be further improved. We agreed with the reviewer that improved estimation models (e.g., 3FLD and improved FLD) could improve the SIF retrieval accuracy as well. To reflect these potential hardware and software improvement methods, the text was revised as follow.
“In addition to the hardware improvement, other SIF retrieval methods (e.g., 3FLD and improved FLD) can be used to increase the retrieval accuracy.”
Q24: P. 17-18, lines 465-475: Using of preliminary estimated SIFm (e.g. at action of different stressors) can be not effective for investigation of group of plants with different values of stressor-induced changes in photosynthetic processes. It is very probable that action of stressor in field conditions will induce different values of stress responses in different plants.
Response:
We understand the reviewer’s concern. In theory, we have the maximum control of inducing stress severity in a controlled environment so SIFm for different stress levels can be obtained to establish a comprehensive model for field applications. We also acknowledge that this is a hypothetic concept and must be tested in future studies.
Q25: Additional general point: text of the manuscript seems to be chaotic; there are small errors (e.g. P. 18, lines 486-492: Author Contributions was not completed). I suppose that additional check of the manuscript can improve it.
Response:
It seems that there is a version control issue, but we have double-checked the revised manuscript without those issues mentioned by the reviewer.

Reviewer 3 Report
In this paper, the authors introduced a ground mobile sensing system to monitoring the photosynthesis system of plants in the field. The measurement results were validated using a PAM system. According to the different measurement methods, the new approach was proven to be promise. This study will help scientist to improve the efficiency for the photo system monitoring of crops, especially in the field.
The experiment design is well. But I have one main question. Due to the manuscript, it seems that the authors only collected the data for once from 24 to 48 HAT. A lot of data from hyperspectral camera and PAM system were collected. As there so many factors involved in this study, how was that be done within one day? Besides, in the text describing PAM measurement, the measuring time and replications were missed. The authors should clarify that in the text.
Specific comments:
L167-168 and Figure 3: The measuring time 0800, 1000, 1200, 1400, 1600 and 1800h made me confusing. I would suggest the authors to revise that to 08:00, 10:00, ...
Figure 8: This figure is quite hard to be read.
Author Response
Responses to Comments from Reviewer #3
Q1: The experiment design is well. But I have one main question. Due to the manuscript, it seems that the authors only collected the data for once from 24 to 48 HAT. A lot of data from hyperspectral camera and PAM system were collected. As there so many factors involved in this study, how was that be done within one day? Besides, in the text describing PAM measurement, the measuring time and replications were missed. The authors should clarify that in the text.
Responses:
Yes, all data presented in the current manuscript were collected in one day. There are several factors to be carefully considered in such experiment design process and data collection arrangement. Firstly, based on an estimation of data acquisition throughput, we designed this experiment with proper number of cultivars, treatments, and replications, so that the field can be scanned within a short time of period (approximately 20 minutes in the present study). According to historic prediction of day time in the study area, the total data collection trials (6 per day) and time intervals (2 hours) were determined. Testing data acquisition trials were conducted several times to ensure that we can collect needed datasets in a day. Secondly, we monitored weather forecast to find the best 2-day time window (partially cloudy and sunny) for this experiment: chemical spray was conducted on Day 1 and data were collected on Day 2. A team effort was required to complete such an intensive data collection for this experiment. We also acknowledge that a single experiment would have many uncertainties and repeated experiments are needed in the future.
Based on the reviewer’s comment, we added the description of measuring time and replications for PAM data.
“Active chlorophyll fluorescence measurements were conducted simultaneously with hyperspectral imaging collection. A human operator followed the GPhenoVision system and used a portable pulse-amplitude-modulation (PAM) fluorometer (OS5p+, Opti-Sciences, Inc., Hudson, NH, USA) to measure the uppermost fully expanded leaf at approximately the fourth mainstem node below the plant terminal. Three leaves per plot were measured.”
Q2: L167-168 and Figure 3: The measuring time 0800, 1000, 1200, 1400, 1600 and 1800h made me confusing. I would suggest the authors to revise that to 08:00, 10:00, …
Responses:
Based on the reviewer’s suggestion, the timestamps were revised to 08:00, 10:00, 12:00, 14:00, 16:00, and 18:00, respectively, in Figure 3.
Q3: Figure 8: This figure is quite hard to be read.
Responses:
This figure contains three components delivering ample information of correlation analysis results. The upper triangle provides correlation values between traits indicated by a row and a column, whereas the lower triangle provides scattering plots between traits. The main diagonal provides distribution subplots for individual traits. Although efforts are needed to read this figure, readers can directly get both qualitative (plots) and quantitative (numeric values) information. Thus, the format is preferred for Figure 8. On the other hand, figure caption was revised to enhance the figure readability.
“Figure 8. Pearson correlation values between growth traits and standardized ETR values estimated using four approaches. Values in the upper triangle were correlation values between traits indicated by a row and a column. Subplots along the main diagonal were the value distribution of individual traits, whereas subplots in the lower triangle were scattering plots between traits indicated by a row and a column. Significant (p-value < 0.05) correlation values were rendered by red color or otherwise by black color.”
